# TRINITY: AN EVOLVED LLM COORDINATOR

**Jinglue Xu**[1*], **Qi Sun**[1,3*], **Peter Schwendeman**[2†],
**Stefan Nielsen**[1], **Edoardo Cetin**[1], **Yujin Tang**[1]
[1] Sakana AI, Japan    [2] University of Michigan, USA    [3] Institute of Science Tokyo, Japan

## ABSTRACT

Combining diverse foundation models is promising, but weight-merging is limited by mismatched architectures and closed APIs. TRINITY addresses this with a lightweight coordinator that orchestrates collaboration among large language models (LLMs). The coordinator, comprising a compact language model ($\approx 0.6$B parameters) and a lightweight head ($\approx 10$K parameters), is optimized with an evolutionary strategy for efficient and adaptive delegation. TRINITY processes queries over multiple turns, where at each turn the coordinator assigns one of three roles (*Thinker*, *Worker*, or *Verifier*) to a selected LLM, effectively offloading complex skill acquisition from the coordinator itself. Extensive experiments demonstrate that TRINITY consistently outperforms individual models and existing methods in various tasks, including coding, math, reasoning, and domain knowledge, while robustly generalizing to out-of-distribution tasks. On established benchmarks, TRINITY achieves state-of-the-art performance, including a new record of 86.2% on LiveCodeBench. Theoretical and empirical analyses highlight two key factors driving this success: (1) the coordinator's hidden-state representations provide rich contextualization of inputs, and (2) under high dimensionality and strict budget constraints, the separable Covariance Matrix Adaptation Evolution Strategy algorithm provides substantial advantages over RL, imitation learning, and random search, leveraging potential block-$\varepsilon$-separability.

## 1 INTRODUCTION

A prominent line of work involving large language models (LLMs) aspires to scale in line with empirical scaling laws, targeting gains by enlarging model size, training tokens, and compute (Kaplan et al., 2020; Hoffmann et al., 2022). Yet the extent to which such scaling remains efficient and yields sustained returns is uncertain and often resource intensive. An alternative at the micro level is model merging (Akiba et al., 2025; Wortsman et al., 2022; Yang et al., 2024; Kuroki et al., 2024), which seeks parameter-level integration. However, this approach is frequently impractical due to architectural incompatibilities and the closed-source nature of many high-performing models. In light of these limitations, we adopt a *macro-level* approach: test-time model composition via coordination, which fuses the complementary strengths of multiple state-of-the-art models from diverse providers without modifying their weights. Leveraging prior data and training investments, this coordination can deliver performance improvements without retraining individual models.

The central challenge for such a coordinator is to acquire a rich contextual understanding of a given query to make an effective decision. We posit that this signal can be efficiently extracted from the internal representation of a compact language model, specifically, its hidden states (Allen-Zhu & Li, 2023). In a self-attention-based transformer model, hidden states encode contextual representations of the input (and, after generation, the output) sequence. Hidden states extracted from inputs alone reflect input context, and those taken post-generation additionally capture the model's produced output and latent reasoning. For output sequences, the penultimate token's hidden state carries rich context. It attends over the entire sequence and guides the prediction of a special token (such as `<\think>` or the EOS token), ensuring a stable output distribution. This leads to our central hypothesis that contextual representations from a small language model (SLM) contain sufficient

---

[*]Equal contribution
[†]Work done during internship at Sakana AI
   Correspondence to: jingluexu@sakana.ai, qisun@sakana.ai, yujintang@sakana.ai

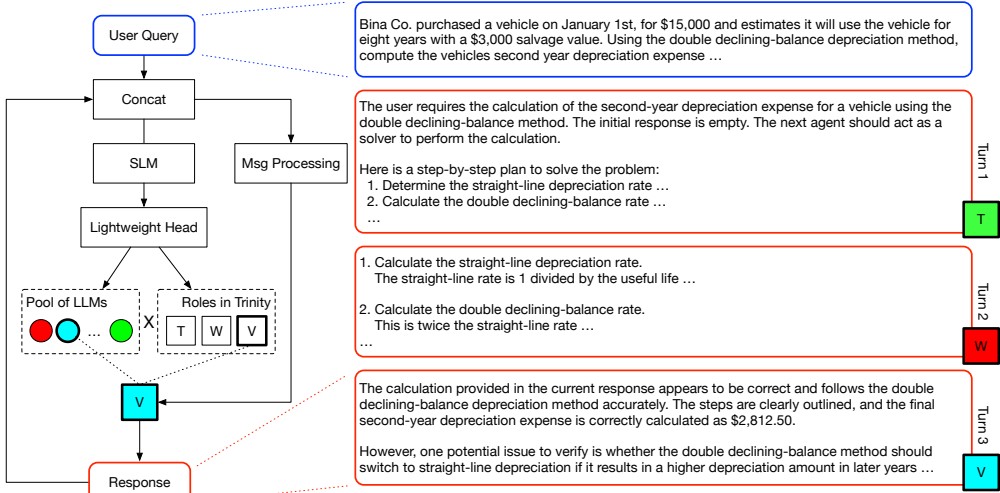

Figure 1: **Overview and an example of our coordination method. Left:** The cyclical coordination architecture. In each turn, the full conversation transcript is passed to a compact coordinator model. A lightweight head selects an LLM and assigns it one of three roles: Thinker (T), Worker (W), or Verifier (V). A message processing module injects a role-specific prompt before the request is sent to the chosen LLM. **Right:** An example of multi-turn coordination. To solve a complex depreciation problem, TRINITY invokes a Thinker (Turn 1) to decompose the task, a Worker (Turn 2) to perform the calculation, and a Verifier (Turn 3) to validate the answer and identify edge cases.

semantic signal for a lightweight head to coordinate multiple LLMs effectively, a possibility that remains underexplored in existing works (see Section 5).

Given these contextual representations, our method, TRINITY, employs an SLM (0.6B parameters) with a lightweight head to orchestrate multiple LLMs (both open- and closed-source models) in a multi-turn protocol, with the total number of learnable parameters under 20K. At each turn, TRINITY selects an LLM and constructs its input by concatenating the original query with the full transcript of prior turns. To ensure the coordinator remains lightweight and offloads complex skill acquisition, TRINITY assigns the selected agent one of three distinct roles: (1) a *thinker* to devise high-level strategies and decompositions; (2) a *worker* to perform concrete problem-solving steps; and (3) a *verifier* to evaluate the current solution's soundness and completeness. The process halts when the verifier is selected and accepts current response as the final answer, or when a fixed-turn budget is exhausted. Figure 1 gives an overview of our method, together with an example of our coordination.

Optimizing this representation-to-coordination mapping is challenging. We observe weak coupling among parameters — each has only a tiny influence on the scalar reward, making traditional methods like REINFORCE's per-parameter gradients low-SNR and therefore ineffective. Training is further constrained by cost, since each step requires running the coordinated agents for inference. We find that a derivative-free Covariance Matrix Adaptation Evolution Strategy (CMA-ES) (Hansen et al., 2003) with diagonal covariance, separable CMA-ES (sep-CMA-ES) (Ros & Hansen, 2008), is effective in this particular regime: high dimensionality, weak parameter correlations, and high per-step cost. We provide theoretical and empirical evidence that, in this extremely budget-tight scenario (1.5k–40k evaluations for a 10k-dimensional problem), sep-CMA-ES significantly outperforms RL and the random search baseline, suggesting strong block-$\varepsilon$-separability (see Definition 1) in the optimization objective.

Across four in-distribution benchmarks including Math500 (Lightman et al., 2023), MMLU (Hendrycks et al., 2020), RLPR (Yu et al., 2025), and LiveCodeBench (Jain et al., 2024), TRINITY consistently outperforms prior methods, achieving a mean relative error reduction of 21.9% over the second-best approach. It also outperforms all single-model baselines with fair, adjusted output-token budgets. Remarkably, TRINITY sets a new state-of-the-art on LiveCodeBench (Jan - Aprl 2025), achieving a pass@1 of $86.2 \pm 0.5\%$. Furthermore, TRINITY is able to zero-shot transfer to four unseen tasks consisting AIME (Veeraboina, 2023), BigCodeBench (Zhuo et al., 2024), MT-Bench (Bai et al., 2024), and GPQA-D (Rein et al., 2024), with performance surpassing each of the single models it orchestrates.

Our main contributions are summarized as follows:

- **A lightweight and effective coordination mechanism.** We show that rich contextual signals from the hidden states of an SLM are sufficient for a tiny head to coordinate multiple diverse LLMs (with the total number of learnable parameters under 20K), a previously underexplored approach to model composition.
- **A highly efficient training methodology.** We demonstrate theoretically and empirically that under the challenging, budget-constrained conditions of our problem, sep-CMA-ES is a superior optimization choice over RL, imitation learning, and random search.
- **State-of-the-art performance and generalization.** TRINITY sets a new record on Live-CodeBench and outperforms existing methods on a wide range of benchmarks. It also generalizes robustly to unseen tasks and develops emergent, task-aware coordination strategies.

## 2 PROBLEM FORMULATION

Let $\mathcal{S}$ be the set of interaction states $s$ (the original query together with the full multi-turn conversation so far). An SLM maps each $s$ to a *representation state* $h(s) \in \mathcal{H} \subset \mathbb{R}^d$ (e.g., a penultimate-token hidden vector). A lightweight coordination head with parameters $\theta \in \mathcal{P} \subset \mathbb{R}^n$ takes $h(s)$ as input and outputs logits over a finite action set $\mathcal{A}$ of agent–role pairs:

$$f_\theta : \ \mathcal{H} \to \mathbb{R}^{|\mathcal{A}|}, \qquad \pi_\theta(a \mid s) \ \propto \ \exp\big(f_\theta(h(s))_a\big), \ a \in \mathcal{A}.$$

The policy $\pi_\theta$ induces a distribution over *all multi-turn trajectories* $\mathcal{T}$, where a trajectory is $\tau = (s_0, a_0, \ldots, s_T)$ with horizon $T \leq B_{\mathrm{turn}}$, where $B_{\mathrm{turn}}$ denotes a fixed turn budget. A terminal reward $R(\tau) \in \{0, 1\}$ is revealed at the end. The optimization objective

$$J(\theta) \ := \ \mathbb{E}_{\tau \sim \pi_\theta}[R(\tau)]$$

is the expected terminal reward of the *coordinator* $\theta$. In short, the representation space $\mathcal{H}$ provides contextual features, while the *coordination space* $\mathcal{P}$ parametrizes policies over *all* trajectories in $\mathcal{T}$. We regard each single, complete, end-to-end run (i.e., sampling of a trajectory $\tau$) as an atomic evaluation, or a Bernoulli call since the rewards follow the Bernoulli distribution. And since each run involves multiple LLM calls, which is a cost we wish to constrain, we seek $\theta^\star \in \arg\max_{\theta \in \mathcal{P}} J(\theta)$ under a tight *atomic evaluation budget* $B_{\mathrm{env}}$ that counts individual Bernoulli calls of the terminal reward used when estimating $J(\theta)$ (e.g., via replication/averaging).

## 3 TRINITY

To address the problem outlined in Section 2, we propose TRINITY, a lightweight and adaptive framework for coordinating multiple diverse LLMs (Figure 1, left). At its core, our approach introduces a coordinator, optimized via sep-CMA-ES, that learns to orchestrate a pool of external LLMs and assign them distinct roles throughout a multi-turn reasoning process.

### 3.1 EFFICIENT PARAMETRIZATION

To efficiently derive the representation and coordination space, the coordinator employs a highly efficient parametrization scheme, as illustrated in Figure 2. We use a pre-trained SLM as a backbone and introduce two distinct sets of trainable parameters.

First, we append a lightweight head directly after the coordinator SLM's final hidden layer. To coordinate $L$ agents, this head projects a hidden state $h \in \mathbb{R}^d$ to an output of size $L + 3$, which provides two sets of logits: $L$ logits for selecting an LLM and three logits for assigning its role. This head defines the fundamental structure of the coordination space. Second, inspired by recent work in efficient fine-tuning (Sun et al., 2025), we adapt a small set of the backbone's layers using a singular value fine-tuning approach. For a selected subset of the coordinator SLM's weight matrices, we perform a singular value decomposition and only learn the singular value scales, keeping the orthogonal matrices fixed. This parameterization scheme is highly efficient, keeping the total number of learnable parameters below 20K, orders of magnitude smaller than typical fine-tuning, while still yielding representational benefits (Figure 5).

Crucially, our method only relies on the head's logit outputs, and the coordinator's generated text is discarded because the job of prompting is delegated to the LLMs in the pool (Section 3.2). Rather than waiting for a full generation, this allows the coordinator to take hidden states corresponding to an earlier token instead of the penultimate to make a quick decision. This combination of extreme parameter efficiency and the potential to make rapid inference makes training the entire TRINITY system with evolutionary strategies uniquely feasible (Section 3.3), avoiding the significant data and computational overhead of imitation learning or RL.

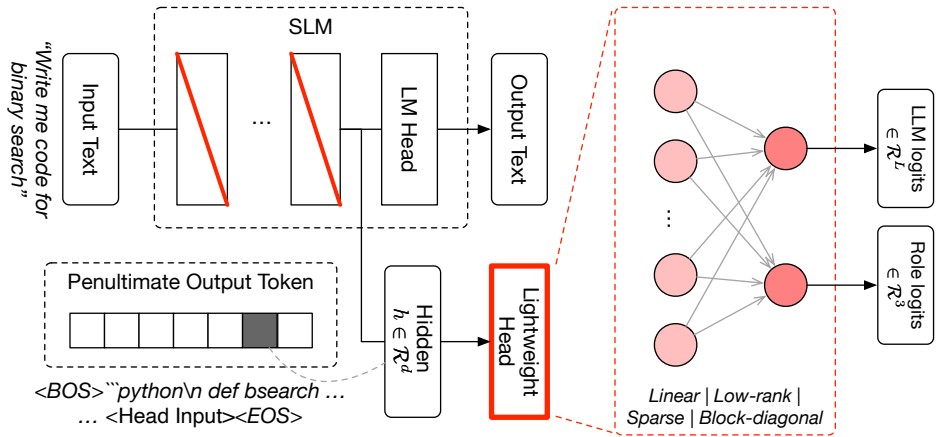

Figure 2: **Parametrization of the TRINITY coordinator.** A lightweight head (see Appendix A.4) operates in parallel to the base model's LM head. It takes the hidden state $h$ corresponding to the penultimate output token as its sole input. This head $f_\theta$ is responsible for coordination decisions, producing two sets of logits, one to select an LLM from the pool of $L$ models, and another to assign one of three roles. We also fine-tune the singular value scales of the parameter matrices in the SLM's layers, indicated by the red diagonal lines. In the figure, the hidden state at the position marked by "<Head Input>" is the input to lightweight head. Note that the semantic correspondence of the decoded message "<BOS> ..." to the hidden state is only for illustrative purpose, as the lightweight head operates on the internal hidden state from that position, not the final decoded text.

## 3.2 TRI-ROLE COORDINATION

Next, we discuss the set of multi-agent interaction patterns available to the coordinator, which are the remaining constructs that define the coordination space.

A key principle of our approach is that the coordinator itself need not be as capable as the underlying agents, its primary function is to *leverage* and *orchestrate* diverse LLMs. Coordination proceeds over at most $K$ turns for a given user query $Q$. Let the transcript after $k-1$ turns be $\mathcal{C}_{k-1} = (Q, O_1, \ldots, O_{k-1})$. At turn $k$, the coordinator selects an agent (i.e., an LLM) $A_k$ from the pool $\mathcal{M}$ and a role $R_k \in \{\text{Thinker (T), Worker (W), Verifier (V)}\}$. The coordinator then prepares a brief, role-specific prompt based on $\mathcal{C}_{k-1}$, queries $A_k$ to obtain a message $M_k$, and lightly post-processes $M_k$ into $O_k$, which is appended to the transcript for the next turn.

In TRINITY, we define three roles, namely *Thinker*, *Worker*, and *Verifier*, each of which enforces a distinct contract between the coordinator and the selected LLM:

- **Thinker strategizes.** The thinker analyzes the current state and returns meta-level guidance, including high-level plans, decompositions, or critiques of partial solutions. Formally, it may propose a plan over subgoals, which the coordinator condenses into $O_k$ to steer subsequent turns, it can also specify the role of the next agent along with the plan.

- **Worker executes.** The worker acts directly on the task to make concrete progress toward a final solution. Given $\mathcal{C}_{k-1}$, it produces actionable content (e.g., a derivation, code snippet, or numerical result). The coordinator extracts the key information and stores it as $O_k$.

- **Verifier evaluates.** The verifier checks whether the accumulated solution in $\mathcal{C}_{k-1}$ is correct, complete, and responsive to $Q$. It outputs a judgment $u_k \in \{\text{ACCEPT}, \text{REVISE}\}$ and an optional diagnosis $\delta_k$. The coordinator records $(u_k, \delta_k)$ as $O_k$ and, if $u_k = \text{ACCEPT}$, signals termination.

The termination time is $\tau = \min\{k \leq K : R_k = \mathrm{V} \text{ and } u_k = \texttt{ACCEPT}\}$, with $\tau = K$ if no acceptance occurs. The final answer returned to the user is $O_\tau$. This rule provides a simple, verifiable stopping condition while allowing the coordinator to allocate the compute budget adaptively across planning, execution, and quality control. See Figure 1 (right) for an example.

### 3.3 LEARNING WITH AN EVOLUTIONARY STRATEGY

To determine a suitable training algorithm, we examine the structure of our problem objective. By varying the head architecture (see Appendix A.4), we observe that the head *block-diagonal-10* retains a large fraction of the performance despite its tiny parameter count (see Section 4.7). These observations reveal that the optimization problem defined in Section 2, when embodied in our representation and coordination space, exhibits strong block-$\varepsilon$ separability (Definition 1). This geometry strongly favors diagonal methods: most of the informative signal is concentrated within blocks, while inter-block interference remains negligible. Conversely, this geometry undermines the REINFORCE baseline (as shown in Section 4.8): noisy global returns swamp weak inter-block signals, yielding ill-conditioned gradients, poor credit assignment, and unstable learning.

We therefore adopt sep-CMA-ES, a black-box evolutionary strategy that iteratively improves a central "parent" policy by sampling a population of perturbed parameter vectors, evaluating each candidate to obtain a fitness score, and recombining candidates via fitness-weighted averaging to form the next parent. Unlike full CMA-ES, sep-CMA-ES maintains only a diagonal covariance matrix, making the algorithm especially well suited to block-diagonal landscapes.

In Appendix A.1, we provide a theoretical analysis tailored to our specific problem regime: a coordination head with about 10K parameters, tight evaluation budgets, binary terminal rewards, and weak but nonzero cross-block couplings. In the following, we present a short comprehensive summary.

Let $n$ be the head dimension, $\lambda = \lceil 4 + 3 \ln n \rceil$ be the CMA-ES population size, and $m_{\mathrm{CMA}}/m_{\mathrm{RS}}$ be the replication counts (number of evaluations per candidate). Denote $T$ as the optimization iteration count. Then, for the small-$T$ regime, **Proposition 1** shows that sep-CMA-ES's improvement grows roughly linearly with the number of iterations, while random search (RS) grows only with the logarithm of how many candidates it can test. Thus, for modest $T$, sep-CMA-ES outperforms RS. In the specific regime of our study ($n \approx 10000$, $\lambda \approx 32$, $m_{\mathrm{CMA}} = 16$, $m_{\mathrm{RS}} = 32$), budget matching yields about $16T$ RS candidates; the gain ratio behaves like $\frac{T}{\ln(16T)} \cdot \eta^2$, where $\eta$ is a reliability factor between 0 and 1, usually close to one. This ratio is greater than one even for small $T$.

**Proposition 2** states that after about $n$ iterations of calibration, sep-CMA-ES enters a steady regime where each step reduces the remaining error by a fraction of order $1/n$, with a rate constant close to $\bar{\kappa}_{\mu,\lambda}$, where the constant $\bar{\kappa}_{\mu,\lambda} = \Theta(1)$ denotes the CMA recombination efficiency. By contrast, RS continues to gain only logarithmically even with repeated rounds. Hence, as $T$ increases, sep-CMA-ES becomes better and the gap compared to RS grows wider.

## 4 EXPERIMENTS

We demonstrate the effectiveness of TRINITY through three key dimensions. First, we directly compare it against both multi- and single- agent baselines in controlled settings. We also show that TRINITY establishes a state-of-the-art performance on the LiveCodeBench task. We then evaluate its generalization capabilities across a diverse set of unseen tasks. Finally, we present analytical results, including ablations and the contextual information encoded in the extracted hidden states, and compare our evolution-based approach against RL, imitation learning, and RS.

### 4.1 EXPERIMENTAL SETUP

**Coordinator and agents.** We use Qwen3-0.6B (Yang et al., 2025) as the coordinator's SLM, paired with a single linear layer of 10K parameters as the simple but effective head, and select the second-to-last layer of the 0.6B model for singular value fine-tuning. Table 6 reports the parameter counts of the various head architectures and the number of parameters updated during singular value fine-tuning. Our model pool contains seven models from both open-source communities and closed-source API providers. These are, three top-tier closed-source models currently available (GPT-5 (OpenAI, 2025), Gemini-2.5-pro (Comanici et al., 2025), and Claude-4-Sonnet (Anthropic,

2025)), and four well-known open-source models (Gemma-3-27B-It (Team et al., 2025), DeepSeek-R1-Distill-Qwen-32B (Guo et al., 2025), Qwen-3-32B (direct), and Qwen-3-32B (reasoning)). Our LLM and training task selection principle is detailed in Appendix A.6.

**Tasks and protocols.** We train and evaluate TRINITY across four diverse tasks, including MATH500, MMLU, RLPR, and LiveCodeBench. For each task, we train on the designated training set and assess performance on the corresponding test set, utilizing official splits where available. For LiveCodeBench specifically, we use the V1 release (400 samples) for training and conduct evaluation on the newly introduced questions in the V6 release (175 samples). To ensure consistency between open and closed models and facilitate training, we set the default maximum generated tokens to 4096 for each LLM, with minimal reasoning effort. We also set the maximum number of coordination turns to five. For assessing generalization capabilities, we further evaluate our approaches on four challenging held-out tasks (AIME2025, BigCodeBench, MT-Bench, and GPQA-D), spanning diverse domains and problem types.

**Baselines.** We compare TRINITY against several categories of baselines. For multi-agent routing methods, we compare against state-of-the-art approaches including MasRouter (Yue et al., 2025), RouterDC (Chen et al., 2024), Smoothie (Guha et al., 2024), MoA (Wang et al., 2024) and random agent selection. We also evaluate individual LLMs in our pool (GPT-5, Gemini-2.5-pro, Claude-4-Sonnet) at both 4K and 20K (marked as 5x CTX) inference tokens to assess performance under accumulated inference budget, and single agent self-reflection over five turns (5x SR). In addition, we include a majority-voting baseline at 5 samples and a baseline with an LLM as the coordinator (see Appendix A.7.3). Detailed experimental settings are provided in Appendix A.7.1.

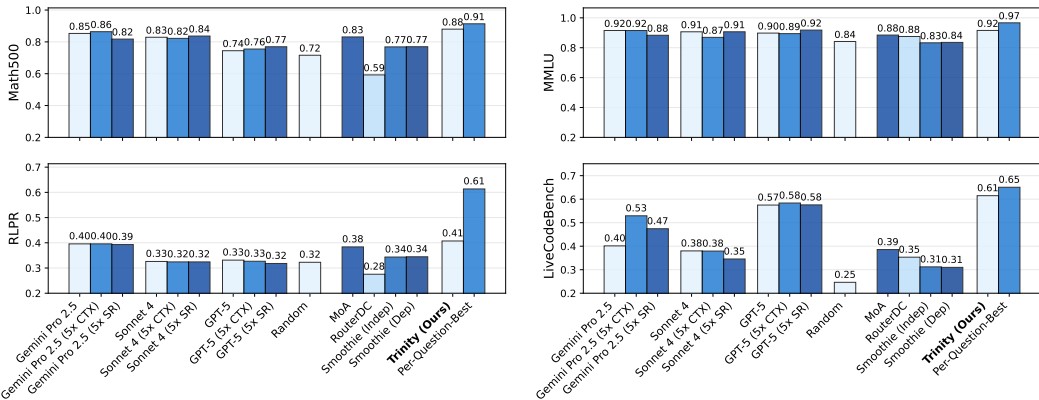

Figure 3: **TRINITY outperforms single- and multi-model baselines across four benchmarks.** Our approach (boldface on the x-axis) achieves the highest performance across four tasks, surpassing the baseline methods. In Math500, MMLU and LiveCodeBench, our performance is close to "Per-Question-Best", representing an upper bound achieved by taking the union of all correct answers from the single LLMs.

## 4.2 IN-DISTRIBUTION EVALUATION

As shown in Figure 3, TRINITY consistently outperforms existing multi-agents methods across all four benchmarks, demonstrating its superior ability to harness the strengths of a diverse LLM pool. While some baseline methods achieve moderate performance on individual tasks, such as MoA's strong results on Math500 (0.83) and RLPR (0.38), they fail to maintain *consistency* across tasks, as evidenced by its relatively weaker performance on LiveCodeBench (0.39). This inconsistency highlights the difficulty of effectively coordinating diverse agents. Notably, some collaboration approaches even degrade performance below random baselines, as seen with Router DC's RLPR score of 0.28 compared to random selection's 0.32, further emphasizing the challenge.

In contrast, TRINITY achieves robustly high performance across the board, including a remarkable 0.61 pass@1 score on LiveCodeBench v6, substantially surpassing all competing methods. Also, we achieve a 11.76% relative error reduction on MATH500 compared to the 2nd best method (Gemini Pro 2.5 with 5x CTX). These results suggest that, while diverse agent capabilities offer significant potential, effective collaboration requires sophisticated mechanisms for optimal organizational decisions, which simple or heuristic-based routing approaches cannot easily achieve.

Table 1: **Performance Across Hold-Out Tasks**

| Model | AIME | BigCodeBench | MT-Bench | GPQA-D | Average |
|---|---|---|---|---|---|
| Gemini Pro 2.5 | 46.67 | 35.10 | 9.37 | 75.25 | 52.34 |
| GPT-5 | 46.67 | 33.80 | 9.35 | 72.73 | 51.07 |
| Claude-4-Sonnet | 35.33 | **35.80** | 9.28 | 67.30 | 46.14 |
| Qwen3-32B (reasoning) | 23.33 | 20.90 | 8.99 | 59.09 | 34.44 |
| DeepSeek-R1-Qwen-32B | 30.00 | 24.30 | 8.43 | 51.01 | 35.10 |
| Qwen3-32B (direct) | 20.00 | 23.00 | 9.03 | 54.05 | 33.46 |
| Gemma-3-27B-IT | 20.00 | 20.30 | 8.76 | 33.33 | 21.38 |
| TRINITY (Ours) | **50.00** | **35.80** | **9.60** | **76.82** | **54.21** |

Compared with single-model baselines, TRINITY outperforms every individual model in the pool, even when they are enhanced with either extended inference budget (5x CTX) or self-reflection settings (5x SR). The 5× inference budget matches our maximum turn setting of five, ensuring that comparisons are fair, and in some cases even favorable to the baselines. A closer look at Figure 3 reveals distinct strengths and limitations for each model. For example, Gemini excels on RLPR and MATH500 but shows moderate performance on LiveCodeBench while GPT-5 dominates it. Remarkably, TRINITY achieves optimal performance across all tasks, demonstrating its ability to dynamically leverage each model's strengths and compose them effectively for different challenges. To further contextualize TRINITY's capability, we also include an upper bound ("Per-Question-Best") representing the performance achieved by taking the union of all correct answers from the seven LLMs in the pool. Our method approaches this limit closely on three of four tasks, demonstrating its ability to harness the collective capabilities of the model ensemble. TRINITY also exhibits upper-tier token efficiency compared to other methods, especially coordination methods. (see Appendix A.7.4 for detailed comparison).

### 4.3 ZERO-SHOT TRANSFER TO UNSEEN TASKS

This suggests that TRINITY does more than simply select the best agent for a task. To assess TRINITY's generalization capability, we tested its zero-shot performance on four held-out benchmarks. As summarized in Table 1, TRINITY achieves the highest average score (54.21) and outperforms every individual baseline on each of the four tasks. It secures top performance on AIME (50.00), MT-Bench (9.60) and GPQA-D (76.82), and ties for first on BigCodeBench (35.80). This result highlights a key advantage of our approach. While individual models exhibit specialization strengths and weaknesses (e.g., Gemini Pro 2.5 and GPT-5 perform better on reasoning tasks compared to coding benchmarks, and Claude-4-Sonnet shows relatively balanced performance), TRINITY delivers consistent results across all domains. It effectively synthesizes the capabilities of the entire pool to achieve emergent performance that surpasses any single constituent model. Surprisingly, we find that Qwen3-32B reasoning mode underperforms direct mode on BigCodeBench, this is mostly due to its verbose reasoning, causing formatting failures in certain test cases.

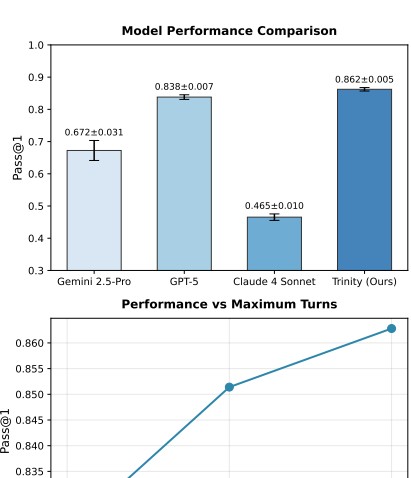

Figure 4: **LiveCodeBench Results**. **Top:** TRINITY achieves state-of-the-art. **Bottom:** TRINITY benefits from increasing maximum turns budgets.

### 4.4 UNLEASHING FULL POWER

Due to hardware constraints in serving open-source models, we limited the maximum output length for all LLMs in the pool for fair comparisons in the previous experiments. For the LiveCodeBench task, the coordinator's LLM selection narrows down to the three closed-models after training. This allows us to remove the output length constraint and observe the full power of TRINITY on LiveCodeBench. Notice that we simply remove the constraints and do not retrain TRINITY.

In Figure 4 (top), TRINITY demonstrates substantial improvements over the constituent models, and achieves state-of-the-art performance with a pass@1 score of 0.862 on LiveCodeBench V6, newly-released questions spanning January to April 2025. This represents a significant improvement

Table 2: **Ablation study results.** We compare the performance on in-distribution tasks when we (1) remove the singular value fine-tuning in SLM; (2) remove the thinker-role selection (3) remove the tri-role selection; and (4) use the last instead of the penultimate token. TRINITY achieves the best overall performance. (5) remove agent selection but keep role selection

| Method | LiveCodeBench | MATH500 | MMLU | RLPR | Average |
|---|---|---|---|---|---|
| TRINITY | **61.46** | **88.00** | 91.56 | 40.72 | **70.44** |
| w/o Singular value fine-tuning | 55.68 | 85.85 | 90.10 | 39.77 | 67.85 |
| w/o Thinker-role selection | 57.80 | 86.20 | **92.75** | 38.00 | 68.69 |
| w/o Tri-role selection | 58.28 | 82.00 | 91.64 | 36.15 | 67.02 |
| w/ Last token | 50.85 | 87.00 | 82.19 | 38.60 | 64.66 |
| Claude-4-Sonnet only | 39.09 | 82.25 | 88.23 | 34.90 | 61.12 |
| Gemini Pro 2.5 only | 46.51 | 83.05 | 79.41 | **43.00** | 62.99 |
| GPT-5 only | 59.54 | 75.66 | 90.74 | 37.87 | 65.95 |

over leading baselines: GPT-5 (0.838), Gemini 2.5-Pro (0.672), and Claude-4-Sonnet (0.465). In addition, Figure 4 (bottom) also shows that TRINITY benefits from increasing max collaboration turns, improving from 0.823 to 0.863 as turns increase from 2 to 6. This pattern makes intuitive sense and implies that TRINITY's capability stems from complex coordination and goes beyond naive routing.

## 4.5 ABLATION STUDIES

We conduct ablation studies to verify the effectiveness of our design choices, with results summarized in Table 2. First, removing the singular value fine-tuning consistently lowers scores, confirming the benefit of adapting the coordinator model's internal representation directly, which allows it to generate more effective signals for the head. Next, gradually removing the tri-role selection —first the thinker role, then the entire tri-role selection— proves detrimental to complex reasoning, causing substantial degradation on MATH500 (-6.0 points) and RLPR (-4.57 points). Additionally, switching to the final token, which often corresponds to a semantically sparse EOS token, causes a severe performance collapse, particularly on LiveCodeBench (more than 10 points drop). Finally, when we remove agent selection and instead send all queries to a single fixed agent while retaining only role selection, performance is significantly undermined. Together, these findings underscore the necessity of the full TRINITY design.

## 4.6 SEPARABILITY IN REPRESENTATION SPACE

The success of our lightweight coordinator depends on a well-structured representation space where hidden states are separable by task. We verify this by extracting hidden states from the coordinator during in-distribution runs and analyzing their linear separability using a suite of methods, including linear classifier (SVM) and dimensionality reduction (t-SNE).

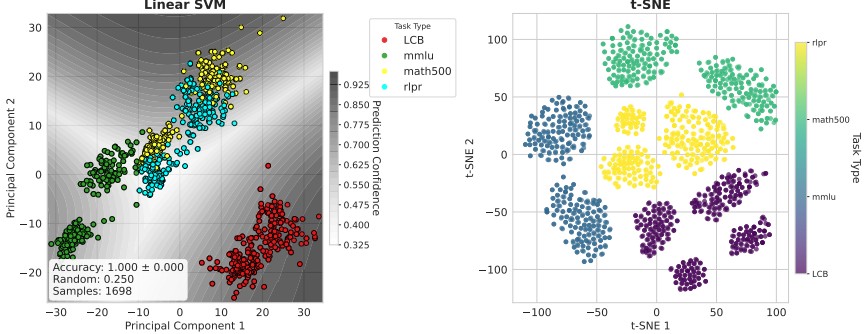

Figure 5: **Task type separability in extracted hidden states.** Both are based on penultimate-token hidden states processed by the SLM on the input sequence, and the labels are from the task metadata.

Appendix A.3 reports the full results, and Figure 5 presents two key analyses of the representation space. A linear SVM achieves perfect classification, far above chance level (0.25 for four classes), indicating near-perfect linear separability. The t-SNE visualization likewise exhibits clear, well-

separated clusters, corroborating strong non-linear separability. This high degree of separability is a key factor that enables our lightweight, linear head to make effective coordination decisions with extreme parameter efficiency. Additional experiments in Appendix A.3 also indicate a positive correlation between the separability in the representation space and the coordinator's performance.

## 4.7 SEPARABILITY IN PROBLEM OBJECTIVE

Changing the head architecture (see Appendix A.4) not only alters coordinator performance, but also reveals structural properties of the problem objective, namely the mapping from hidden states to agent/role choices that maximizes downstream task reward. Table 3 shows that *linear* is the most reliable choice overall across LiveCodeBench, RLPR, Math500, and MMLU, with *sparse* edging it out by a negligible margin on MMLU only. The *block-diagonal-10* head paired with an `argmax` output conversion is intentionally designed to maximize independence among the ten logits—one block per agent/role—thereby suppressing inter-logit correlations. Parameter-count wise, this head uses only $d_h$ weights (about $10\times$ fewer than the *linear*'s $d_h n_a$; e.g., 1,024 vs. 10,240 when $d_h$=1,024, $n_a$=10) and still retains competitive mid-tier performance. Importantly, `argmax` further increases independence by removing the softmax simplex constraint. With `argmax`, decisions depend only on the largest logit, so perturbations to non-maximal blocks neither reduce nor redistribute probability mass, which reduces cross-block interference in both inference and fitness attribution. This result suggests strong block-$\varepsilon$ separability (Definition 1) as a property of the coordination objective, in addition to the geometric separability of hidden states studied in Section 4.6.

Table 3: **Results by varying heads and output conversion.** By default, the output conversion is softmax normalization. For *block-diagonal-10*, the output conversion is argmax.

| Head | LiveCodeBench | MATH500 | MMLU | RLPR |
|------|---------------|---------|------|------|
| *linear* | **0.615** | **0.880** | 0.916 | **0.401** |
| *low-rank* | 0.597 | 0.770 | 0.914 | 0.344 |
| *sparse* | 0.400 | 0.811 | **0.917** | 0.372 |
| *block-diagonal-2* | 0.336 | 0.776 | 0.897 | 0.378 |
| *block-diagonal-10* + argmax | 0.551 | 0.812 | 0.802 | 0.376 |

## 4.8 SEP-CMA-ES VS RANDOM SEARCH VS REINFORCE VS SUPERVISED FINE-TUNING

To empirically demonstrate the advantages of sep-CMA-ES for our setting (Section 2), we compare it against REINFORCE (Williams, 1992), SFT, and RS with fitness averaging, which is appropriate for binary rewards (see Appendix A.5). Table 4 shows that sep-CMA-ES outperforms other algorithms for training the coordinator, consistent with our theory (Section 3.3, Appendix A.1). REINFORCE exhibits jagged, high-variance learning curves with weak overall progress, which is expected under terminal (binary) rewards and weak parameter correlation.

Table 4: **Comparison of sep-CMA-ES with REINFORCE, SFT, and RS.** We compare the performance on in-distribution tasks for four learning algorithms under comparable budgets

| Method | LiveCodeBench | MATH500 | MMLU | RLPR |
|--------|---------------|---------|------|------|
| REINFORCE | 0.253 | 0.459 | 0.500 | 0.266 |
| RS | 0.374 | 0.794 | 0.897 | 0.345 |
| SFT | 0.592 | 0.786 | 0.906 | 0.360 |
| sep-CMA-ES | **0.615** | **0.880** | **0.916** | **0.401** |

As shown in Figure 6, sep-CMA-ES adapts to a meaningful agent selection distribution that favors high-performing LLMs. By contrast, REINFORCE maintains an almost uniform selection pattern, indicating ineffective policy improvement. Although not shown in the figure, RS often collapses to unipolar choices, over-selecting a single agent or role and thereby significantly limiting diversity of agents and roles, which degrades performance. While SFT achieves competitive gains, it does not scale to multi-turn coordination due to the prohibitive cost of label generation (see Appendix A.2).

# 5 RELATED WORKS

We use *Model fusion* to refer to methods that combine multiple models into a more capable system. Prior work divides into two complementary levels: *micro-level* fusion in *parameter space*, where

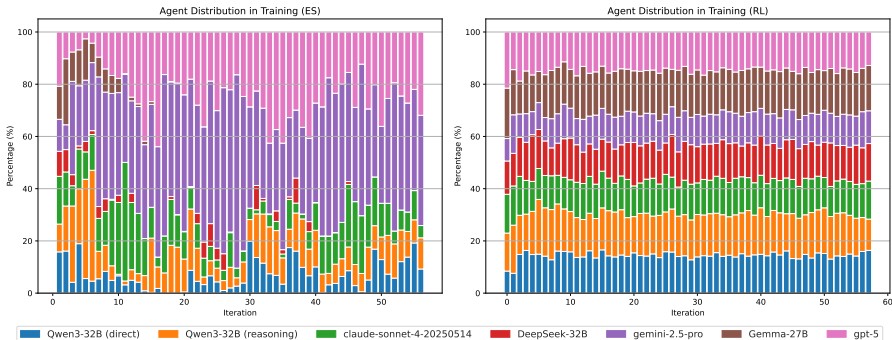

Figure 6: **LLM selection distribution evolves as the coordinator learning progresses. Left:** Distribution evolution from sep-CMA-ES. **Right:** Distribution evolution from REINFORCE.

weights of parent models are merged into a child model, and *macro-level* fusion in *data-flow space*, where activations or outputs are passed across fixed models or model components.

**Micro-level**. Early approaches in micro-level utilize on static recipes such as weight averaging or task-balanced interpolation to integrate multiple model capabilities across domains with minimal computation (Goddard et al., 2024). More recent work has introduced optimization-based methods to model-merging: for example, an evolutionary framework that searches over "merging recipes", demonstrating that learned strategies can outperform hand-designed ones and yield stronger generalization (Akiba et al., 2025). However, because micro-level model-fusion is performed in the parameter-space, these methods face the core limitation that they require access to model weights with compatibility requirements (Yadav et al., 2023; Yu et al., 2024). This confines their applicability to open-source checkpoints, while excluding the closed-source models that currently define the frontier of performance. Consequently, micro-level fusion cannot incorporate the strongest available models, motivating the exploration of data-space approaches that treat models as black boxes.

**Macro-level**. Model fusion in the data-flow space can itself be performed at multiple degrees. Earlier works have allowed propagation of tensors through layers taken from different models (Bansal et al., 2021) or sequentially processing individual tokens by different models (Muqeeth et al., 2024). Our work most relates to a broader view of macro-level model fusion in which methods create stronger singular models by scaffolding or routing between multiple agents. In particular, multi-agent scaffolding techniques like Mixture of Agents (MoA) (Wang et al., 2024) and Multi-Agent Debate (MAD) (Liang et al., 2023) form networks of agents which can extract capabilities from each individual model. Routing methods, such as Smoothie (Guha et al., 2024) or RouterDC (Chen et al., 2024) aim to choose the best model or model response for a given question. Similarly, Mas-Router (Yue et al., 2025) combines both by routing agents and human-designed scaffolds to form an adaptive multi-agent model per question. These methods rely on expensive multi-model inference or static, human-designed collaboration patterns. In contrast, TRINITY introduces a lightweight, learned coordinator that adaptively assigns dynamic roles to LLMs, utilizing the contextual representation generated from a SLM.

## 6 CONCLUSIONS

In this work, we introduce TRINITY, a framework demonstrating that a lightweight coordinator can orchestrate diverse LLMs to achieve state-of-the-art performance. Leveraging a tri-role protocol and trained with a highly efficient evolutionary strategy, our results suggest a promising path forward lies in engineering collaborative AI ecosystems rather than scaling monolithic models. A key limitation, however, is the gap between abstract reasoning and grounded execution, as the system can devise plans involving tools but cannot yet act on them. Future work will therefore focus on integrating a more heterogeneous pool of agents, including code interpreters and APIs, to bridge this gap and create a more general and capable problem-solving system.

ACKNOWLEDGEMENTS

We thank Koshi Eguchi and Kou Misaki for the infrastructure support, and the entire Sakana AI R&D team for their valuable comments and suggestions.

AUTHORS CONTRIBUTIONS

Jinglue Xu designed and curated the training datasets, conducted both theoretical and empirical analyses, and led the subsequent algorithmic and implementation development. Qi Sun proposed, named, and implemented the role selection algorithm, and led the training and evaluation experiments. Stefan Nielsen designed training and evaluation configurations, implemented baselines, and contributed to shaping the trinity roles. Edoardo Cetin contributed to shaping early algorithmic designs and advised the project. Yujin Tang initiated and led the project, implemented the initial algorithm, and conducted the first experiments. All authors contributed to the experimental design and paper writing.

**Ethics statement.** Our approach focuses on collaboration between agents to achieve better performance on existing benchmarks. As this work involves only computational improvements to established evaluation tasks without involving human subjects, sensitive data, or potential misuse applications, we identify no ethical concerns.

**Reproducibility statement.** To ensure full reproducibility of our results, we provide comprehensive resources in the supplementary material, including source code and trained model weights. We also detail all model and task selection decisions within the paper. All base models and datasets used in this work are publicly available.

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

## A  APPENDIX

### A.1  THEORETICAL ANALYSIS OF SEP-CMA-ES

In this section, we compare sep-CMA-ES with random search (RS) for maximizing $J$ over $\mathcal{P}$ under binary rewards and strict budgets. All analyses are carried out in a covariance-normalized chart and mapped back through the current diagonal $D_t$, fixing the metric mismatch between selection (in whitened coordinates tied to $\mathcal{P}$) and stepping (in the original coordinates of $\mathcal{P}$). A Hessian-based block-$\varepsilon$ separability condition on $g := -J$ controls inter-block couplings after a positive diagonal scaling and links to the algorithm's dynamic scaling via a diagonal-comparability assumption. Concentration bounds translate replication $m$ into rank-quality attenuation without moment swapping. We also instantiate the specific case in our study ($n \approx 10000$, $m_{\mathrm{RS}} = 32$, $m_{\mathrm{CMA}} = 16$), directly tying the representation $h(s) \in \mathcal{H}$ and the coordinator parameters $\theta \in \mathcal{P}$ to the observed efficiency of sep-CMA-ES.

#### A.1.1  DEFINITIONS AND ASSUMPTIONS

**Notations.**  We optimize $J(\theta) = \mathbb{E}[R(\tau)]$ over the *coordination space* $\mathcal{P} \subset \mathbb{R}^n$ induced by the head $f_\theta : \mathcal{H} \to \mathbb{R}^{|\mathcal{A}|}$ acting on *representation states* $h(s) \in \mathcal{H} \subset \mathbb{R}^d$. Set $g(\theta) := -J(\theta)$ and $H(\theta) := \nabla^2 g(\theta)$. We analyze contraction toward the origin (w.l.o.g. by re-centering $\mathcal{P}$) in a compact domain $\mathcal{D} \subset \mathcal{P}$. sep-CMA-ES maintains mean (iterate) $m_t \in \mathcal{P}$ with radius $r_t := \|m_t\|$, step-size $\sigma_t > 0$, and diagonal scaling

$$D_t = \mathrm{diag}(\sqrt{s_{1,t}}, \ldots, \sqrt{s_{n,t}}) \succ 0, \qquad y = m_t + \sigma_t D_t z, \ \ z \sim \mathcal{N}(0, I_n).$$

Whitened chart: $x = D_t^{-1}(y - m_t) = \sigma_t z$ (isotropic sampling). Direction: $u_t := m_t/\|m_t\|$, projection $Z_\| := \langle u_t, z \rangle \sim \mathcal{N}(0, 1)$. Population size $\lambda = \lceil 4 + 3\ln n \rceil$ ($\geq 2$), $\mu$ parents with weights $(w_j)_{j=1}^\mu$; $z_{j:\lambda}$ are order statistics. RS uses a fixed replication count per candidate; in this appendix we set $m_{\mathrm{RS}} := 32$. The atomic budget $B_{\mathrm{env}}$ counts Bernoulli calls.

**Blocks, scaling, and operators.**  Let $\{B_1, \ldots, B_M\}$ partition $\{1, \ldots, n\}$ (coordinate blocks in $\mathcal{P}$). For any matrix $M$, $\mathrm{off}(M)$ zeroes its diagonal; $\mathrm{off}_{\mathrm{inter}}(M)$ zeroes diagonal and within-block entries. For diagonal $D$, let $s_{\max}(D)$, $s_{\min}(D)$ be its largest/smallest diagonal square-roots and define

$$\kappa_D := \frac{s_{\max}(D)^2}{s_{\min}(D)^2}, \qquad \kappa_D(t) := \frac{s_{\max}(D_t)^2}{s_{\min}(D_t)^2}.$$

**Definition 1 (Hessian-based block-$\varepsilon$ separability in $\mathcal{P}$)**  *There exists a structural diagonal scaling $S = \mathrm{diag}(s_1, \ldots, s_n) \succ 0$ such that the scaled Hessian $H_S(\theta) := S^{1/2} H(\theta) S^{1/2}$ is uniformly nearly block-diagonal on $\mathcal{D}$. With $D(\theta) := \mathrm{diag}(H_S(\theta))$, one of the following dimensionless bounds holds with a common $\varepsilon_H \in [0, 1)$:*

$$\sup_{\theta \in \mathcal{D}} \left\| D(\theta)^{-1/2} \, \mathrm{off}_{\mathrm{inter}}\big(H_S(\theta)\big) \, D(\theta)^{-1/2} \right\|_2 \leq \varepsilon_H, \tag{B1}$$

$$\sup_{\theta \in \mathcal{D}} \max_{\substack{i \in B_p, \, j \in B_q \\ p \neq q}} \frac{|[H_S(\theta)]_{ij}|}{\sqrt{[H_S(\theta)]_{ii}[H_S(\theta)]_{jj}}} \leq \varepsilon_H, \tag{B2}$$

$$\sup_{\theta \in \mathcal{D}} \max_{i \in B_p} \frac{\sum_{j \in B_q, \, q \neq p} |[H_S(\theta)]_{ij}|}{[H_S(\theta)]_{ii}} \leq \varepsilon_H \ (< 1). \tag{B3}$$

*Within-block structure is unrestricted; $0 < \mu_i \leq [H_S(\theta)]_{ii} \leq L_i < \infty$ on $\mathcal{D}$.*

**Assumption 1 (Diagonal comparability)**  *There exist constants $c_{\mathrm{cmp}}, C_{\mathrm{cmp}} > 0$ such that for all $t, i$,*

$$c_{\mathrm{cmp}} \leq \frac{s_i}{s_{i,t}} \leq C_{\mathrm{cmp}}, \qquad \text{equivalently } C_{\mathrm{cmp}}/c_{\mathrm{cmp}} = O\big(\sup_t \kappa_D(t)\big).$$

*This links the structural scaling $S$ in Definition 1 to the algorithm's dynamic scaling $D_t$.*

**Definition 2 (Metric–alignment factor)**  *For any unit $u$ and diagonal $D \succ 0$,*

$$\chi(u, D) := \frac{(u^\top D u)^2}{u^\top D^2 u} \in \left[\frac{1}{\kappa_D}, 1\right],$$

*the squared correlation between the ranking score $\langle u, z \rangle$ (whitened) and the progress score $\langle Du, z \rangle$ (original metric on $\mathcal{P}$).*

**Assumption 2 (Local linear score with curvature remainder)** *There exist $\gamma > 0$, $L_{\mathrm{curv}} \geq 0$, and a step-size window such that along the trajectory (in whitened coordinates)*

$$J(m_t + \sigma D_t z) = \tfrac{1}{2} + \gamma \, \sigma \, \langle u_t, z \rangle + \xi_t(z), \qquad |\xi_t(z)| \leq (L_{\mathrm{curv}} + c_H \varepsilon_H) \, \sigma^2 \|z\|^2,$$

*with constant $c_H > 0$.*

**Definition 3 (Rank attenuation under replication)** *Let $N = \lfloor B_{\mathrm{env}}/m_{\mathrm{RS}} \rfloor$ be the total number of RS candidates evaluated under the budget and $Z_\parallel^\star := \min_{1 \leq k \leq N} \langle u_t, z^{(k)} \rangle$. With $x_- := \min\{x, 0\}$,*

$$\tilde{\rho}_{\mathrm{RS}}^2 := \frac{\mathbb{E}\left[\left(Z_\parallel^{\mathrm{sel}}\right)_-^2\right]}{\mathbb{E}\left[\left(Z_\parallel^\star\right)_-^2\right]} \in [0, 1], \qquad \tilde{\rho}_{\mathrm{CMA}}^2 := \frac{\mathbb{E}\left[\left\langle u_t, \sum_{j=1}^\mu w_j z_{j:\lambda}^{(\widehat{q}_{m_{\mathrm{CMA}}})} \right\rangle^2\right]}{\mathbb{E}\left[\left\langle u_t, \sum_{j=1}^\mu w_j z_{j:\lambda}^{(-Z_\parallel)} \right\rangle^2\right]} \in [0, 1].$$

**Assumption 3 (Metric–alignment comparability)** *There exists $C_\chi \geq 1$ such that for relevant $t$,*

$$\frac{1}{C_\chi} \leq \frac{\chi(u_t, D_t)/\kappa_D(t)}{\chi(u_0, D_0)/\kappa_D(0)} \leq C_\chi.$$

*Thus the alignment efficiency $\chi(u_t, D_t)/\kappa_D(t)$ stays within a bounded factor of its initial value.*

### A.1.2 SEP-CMA-ES VS RANDOM SEARCH WITH FITNESS AVERAGING

**Rank noise and attenuation.** Consider two candidates $z_1, z_2$ in the same batch with linear score gap $\Delta := \gamma \sigma |\langle u_t, z_1 - z_2 \rangle|$. Averaging $m$ Bernoulli draws per candidate yields the misorder bound

$$\Pr\left[\widehat{f}(m_t + \sigma D_t z_1) \leq \widehat{f}(m_t + \sigma D_t z_2) \text{ but } J(m_t + \sigma D_t z_1) > J(m_t + \sigma D_t z_2)\right]$$

$$\leq C e^{-c\,m\,\Delta^2} + \Pr(\mathsf{curv} > \tfrac{\Delta}{2}),$$

where the curvature event $\{\mathsf{curv} > \Delta/2\}$ is due to $\xi_t$ and admits the tail

$$\Pr(\mathsf{curv} > \tfrac{\Delta}{2}) \leq C' \exp\left(-c' \frac{\Delta}{\sigma^2 \varepsilon_H}\right) + O(\varepsilon_H).$$

Hence, for $\sigma$ in a local monotonicity window (Assumption 2) the signal-to-curvature ratio is order $1/\varepsilon_H$, giving an exponential suppression of curvature-induced flips. To scale this pairwise guarantee to batch selection, restrict attention to the $O(\log N)$ (RS) or $O(\log \lambda)$ (CMA) most competitive order statistics: by extreme-value theory, the typical spacing between the winner and the next competitors is $\Theta(1/\sqrt{\ln N})$, and union-bounding only within this top cluster yields

$$\tilde{\rho}_{\mathrm{RS}}^2 \geq 1 - C_1 N \log N \cdot p_{\mathrm{flip}}(m_{\mathrm{RS}}) - C_2 \varepsilon_H, \qquad \tilde{\rho}_{\mathrm{CMA}}^2 \geq 1 - C_1 \lambda \log \lambda \cdot p_{\mathrm{flip}}(m_{\mathrm{CMA}}) - C_2 \varepsilon_H,$$

with $p_{\mathrm{flip}}(m) \lesssim e^{-cm\gamma^2\sigma^2} + e^{-c'/(\varepsilon_H)} + O(\varepsilon_H)$. In particular, choosing

$$m \geq \frac{1}{c\gamma^2\sigma^2}\left(\ln N + \ln\ln N + \ln\tfrac{1}{\delta}\right) \quad \text{or} \quad m \geq \frac{1}{c\gamma^2\sigma^2}\left(\ln\lambda + \ln\ln\lambda + \ln\tfrac{1}{\delta}\right)$$

ensures $\tilde{\rho}^2 \geq 1 - \delta - O(\varepsilon_H)$ for RS or CMA respectively. This gives a direct budget–replication trade-off inside $\mathcal{P}$.

**Budget-normalized single-round RS gain.** Let $Z_\parallel^\star = \min_{1 \leq k \leq N} \langle u_0, z^{(k)} \rangle$ and $v_N^2 := \mathbb{E}[(-Z_\parallel^\star)^2] = 2\ln N + O(\ln\ln N)$. Define the high-probability event controlling batch norms

$$E_N := \left\{\max_{1 \leq k \leq N} \|z^{(k)}\|^2 \leq n + 2\sqrt{nt} + 2t\right\}, \qquad t = \ln(N/c_0),$$

so that $\Pr(E_N) \geq 1 - c_0$ by a Laurent–Massart tail plus a union bound. On $E_N$, the oracle step along $D_0 z^{\mathrm{sel}}$ (with $z^{\mathrm{sel}}$ the noisy-rank-selected candidate) is

$$\sigma^\star = \big( - \frac{\langle m_0, D_0 z^{\mathrm{sel}} \rangle}{\|D_0 z^{\mathrm{sel}}\|^2} \big) \vee 0, \quad r_0^2 - \|m_0 + \sigma^\star D_0 z^{\mathrm{sel}}\|^2 = r_0^2 \frac{\big( \langle u_0, D_0 z^{\mathrm{sel}} \rangle \big)_-^2}{\|D_0 z^{\mathrm{sel}}\|^2}.$$

Because selection is driven by $\langle u_0, z \rangle$ in the whitened chart and geometric progress depends on $\langle D_0 u_0, z \rangle$, the squared correlation factor $\chi(u_0, D_0) = \frac{(u_0^\top D_0 u_0)^2}{u_0^\top D_0^2 u_0} \in [1/\kappa_D, 1]$ appears multiplicatively in the numerator's expectation, while the denominator is controlled by $\kappa_D = s_{\max}(D_0)^2 / s_{\min}(D_0)^2$ on $E_N$. After integrating out the event complement (which contributes $O(\sqrt{c_0 \, \mathbb{E}[(Z_\|^{\mathrm{sel}})_-^4]}) = O((\ln N)^{1/2} N^{-1/2})$), we obtain

$$\frac{r_0^2 - \mathbb{E}[\min_{\sigma \geq 0} \|m_0 + \sigma D_0 z^{\mathrm{sel}}\|^2]}{r_0^2} \; \geq \; \chi(u_0, D_0) \cdot \frac{(1 - \delta_N) \, \tilde{\rho}_{\mathrm{RS}}^2 \, v_N^2}{\kappa_D \big( n + 2\sqrt{n \ln(N/c_0)} + 2 \ln(N/c_0) \big)} - C \varepsilon_H, \tag{1}$$

for a universal $C > 0$ and $\delta_N = O((\ln N)^{1/2} N^{-1/2})$. A fixed $\sigma$ within the local monotonicity window loses only a universal constant factor.

**Per-iteration CMA gain and geometric regime.** Let

$$\alpha_{\mu,\lambda} := \mathbb{E}\left[ \left\langle u_t, \sum_{j=1}^\mu w_j z_{j:\lambda} \right\rangle \right], \quad \beta_{\mu,\lambda} := \mathbb{E}\left[ \Big\| \sum_{j=1}^\mu w_j z_{j:\lambda} \Big\|^2 \right], \quad \kappa_{\mu,\lambda} := \frac{\alpha_{\mu,\lambda}^2}{\beta_{\mu,\lambda}} = \Theta(1/n),$$

and $\bar{\kappa}_{\mu,\lambda} := n \, \kappa_{\mu,\lambda} = \Theta(1)$. The oracle scalar step along $D_t \sum_{j=1}^\mu w_j z_{j:\lambda}$ yields

$$\frac{\mathbb{E}[r_t^2 - r_{t+1}^2]}{r_t^2} \; \geq \; \chi(u_t, D_t) \cdot \frac{1}{\kappa_D(t)} \, \kappa_{\mu,\lambda} \, \tilde{\rho}_{\mathrm{CMA}}^2 \; - \; C \varepsilon_H. \tag{2}$$

The factor $\tilde{\rho}_{\mathrm{CMA}}$ absorbs all rank noise effects (including sign inversions of the recombination direction); $\chi(u_t, D_t)/\kappa_D(t)$ quantifies directional metric mismatch; and the $O(\varepsilon_H)$ term accounts for inter-block perturbations. Under a standard diagonal learning rate $c_{\mathrm{cov}} = \Theta(1/n)$, block-$\varepsilon_H$ separability and diagonal comparability imply that after $T = \Theta(n)$ iterations $D_t$ enters an $O(\varepsilon_H)$-neighborhood of a stationary point, with

$$\mathbb{E}[r_{t+1}^2 \mid r_t] \; \leq \; \Big( 1 - \frac{\bar{\kappa}_{\mu,\lambda}}{n} tilde\rho_{\mathrm{CMA}}^2 (1 - O(\varepsilon_H)) \Big) r_t^2, \tag{3}$$

so the method achieves geometric decay at rate $\Omega(1/n)$ per iteration once stabilized.

**Head-to-head ratio and multi-round RS.** Under a common atomic budget $B_{\mathrm{env}}$, CMA uses $m_{\mathrm{CMA}} \lambda$ evaluations per iteration so $T = \lfloor B_{\mathrm{env}}/(m_{\mathrm{CMA}} \lambda) \rfloor$, while RS evaluates $N = \lfloor B_{\mathrm{env}}/m_{\mathrm{RS}} \rfloor$ candidates. Combining equation 1 and equation 2, and invoking Assumption 3 to cancel $\chi/\kappa_D$ up to a constant, gives

$$\frac{\text{CMA gain}}{\text{RS gain}} \; \gtrsim \; \frac{\bar{\kappa}_{\mu,\lambda}}{2} \cdot \frac{B_{\mathrm{env}}}{m_{\mathrm{CMA}} \lambda} \cdot \frac{n + 2\sqrt{n \ln N} + 2 \ln N}{v_N^2} \cdot \frac{\tilde{\rho}_{\mathrm{CMA}}^2}{\tilde{\rho}_{\mathrm{RS}}^2} - C \varepsilon_H, \quad v_N^2 \sim 2 \ln N. \tag{4}$$

If RS expends its budget across $T$ rounds with fresh batches $N_t$ and fixed (or monotone) $\sigma$ within the window, then gains add roughly as $\sum_t \Theta((\ln N_t)/n)$, which is at most $\Theta((\ln B_{\mathrm{env}})/n)$ for balanced $N_t$—still logarithmic in budget—whereas CMA accumulates *linearly* across iterations (until stabilization), explaining the systematic advantage in budget-tight regimes.

**Trinity-scale instantiation.** For $n \approx 10000$, $\lambda = \lceil 4 + 3 \ln n \rceil = \lceil 4 + 3 \ln 10000 \rceil = 32$. With $m_{\mathrm{CMA}} = 16$ and $m_{\mathrm{RS}} = 32$, budget matching across $T$ CMA iterations yields $N = \lfloor (m_{\mathrm{CMA}} \lambda / m_{\mathrm{RS}}) T \rfloor = \lfloor (16 \cdot 32/32) T \rfloor \approx \lfloor 16 T \rfloor$. This gives $v_N^2 \approx 2 \ln N$. Replication ensures $\tilde{\rho}_{\mathrm{CMA}}^2 \approx 1$ (up to $O(\varepsilon_H)$). Plugging these into equation 4 shows that with the same $B_{\mathrm{env}}$ CMA's gain dominates for modest $T$ (a few to a few dozen iterations), consistent with empirical results where the head acts on $h(s) \in \mathcal{H}$ and updates $\theta \in \mathcal{P}$ under strict budgets.

**Proposition 1** *Fix $T \in [2, 60]$ and let the CMA budget be $B_{\text{env}} = m_{\text{CMA}}\lambda T$. If the replication schedule ensures $\tilde{\rho}_{\text{CMA}}/\tilde{\rho}_{\text{RS}} \geq \eta \in (0, 1]$ and the metric-alignment efficiency stays comparable across iterations (Assumption 3), then, up to an $O(\varepsilon_H)$ term,*

$$\frac{\text{CMA gain in } J}{\text{RS gain in } J} \gtrsim \frac{\bar{\kappa}_{\mu,\lambda}}{2} \cdot \frac{T}{\ln\big(\max\{e, \lfloor (m_{\text{CMA}}\lambda/m_{\text{RS}})\,T\rfloor\}\big)} \cdot \eta^2$$
$$- \frac{C}{\ln\big(\max\{e, \lfloor (m_{\text{CMA}}\lambda/m_{\text{RS}})\,T\rfloor\}\big)}.$$

*The inequality holds for oracle step-sizes and, up to a universal constant factor, for fixed step-sizes within the local monotonicity window (Assumption 2).*

*Proposition 1. Proof.* Set $N = \lfloor (m_{\text{CMA}}\lambda/m_{\text{RS}})T\rfloor$ so both methods consume the same budget. Use equation 1 with $v_N^2 = 2\ln N + O(\ln\ln N)$ and $\delta_N = O((\ln N)^{1/2}N^{-1/2})$ to bound RS improvement. Sum equation 2 over $t = 0, \ldots, T - 1$ to get CMA improvement at least $\sum_t (\chi(u_t, D_t)/\kappa_D(t))\kappa_{\mu,\lambda}\tilde{\rho}_{\text{CMA}}^2 - CT\varepsilon_H$. Apply Assumption 3 to replace iteration-wise factors by a constant multiple; the metric terms cancel in the ratio. Substitute $\kappa_{\mu,\lambda} = \bar{\kappa}_{\mu,\lambda}/n$ and compare $n$ to $v_N^2 \sim 2\ln N$ to obtain the bound with $(\tilde{\rho}_{\text{CMA}}/\tilde{\rho}_{\text{RS}})^2$.

**Proposition 2** *Under Definition 1, Assumptions 1, 2, and 3, and a replication schedule with $\tilde{\rho}_{\text{CMA}}^2 = 1 - O(\varepsilon_H)$, sep-CMA-ES achieves, after a $\Theta(n)$ transient, the per-iteration contraction*

$$\frac{\bar{\kappa}_{\mu,\lambda}}{n}(1 - O(\varepsilon_H)),$$

*i.e., $\mathbb{E}[r_T^2] \lesssim \exp(-c'T/n)\,r_0^2$ for some $c' > 0$ depending on $\bar{\kappa}_{\mu,\lambda}$ and the residual $O(\varepsilon_H)$. Restricting to diagonal covariances incurs only an $O(\varepsilon_H)$ multiplicative loss relative to the block-diagonal optimum.*

*Proposition 2. Proof.* (i) *Scale stabilization:* With $c_{\text{cov}} = \Theta(1/n)$ and block-$\varepsilon_H$ separability plus diagonal comparability, standard CMA drift shows $D_t$ reaches an $O(\varepsilon_H)$-neighborhood of a stationary point in $T_0 = \Theta(n)$ steps; then $\kappa_D(t) = \Theta(1)$ and typical $\chi(u_t, D_t) = \Theta(1)$. (ii) *Uniform per-iteration gain:* Insert these bounds into equation 2 to get $\mathbb{E}[r_{t+1}^2 \mid r_t] \leq (1 - \bar{\kappa}_{\mu,\lambda}\tilde{\rho}_{\text{CMA}}^2/n\,(1 - O(\varepsilon_H)))r_t^2$; iterate to obtain geometric decay with rate $\Omega(1/n)$. (iii) *Closeness to the independent-block ideal:* Since $H_S(\theta)$ is $O(\varepsilon_H)$-close (operator norm) to block-diagonal on $\mathcal{D}$, the population-optimal full-covariance CMA differs from its block-diagonal part by $O(\varepsilon_H)$, so using only diagonals loses $O(\varepsilon_H)$ in the contraction constant. (iv) *Rank reliability:* Replication with $m_{\text{CMA}} \gtrsim (\gamma^2\sigma^2)^{-1}\log\lambda$ keeps $\tilde{\rho}_{\text{CMA}}^2 = 1 - O(\varepsilon_H)$.

## A.2 Supervised fine-tuning

### A.2.1 Experiment details

In this section, we describe our setup and results for experiments with a widely used imitation learning method, SFT. Concretely, we use a direct single-step state–action formulation where each training example consists of a state and a discrete action corresponding to the choice of a single LLM from the pool. SFT trains on these observed state–action pairs to imitate an oracle policy. Given a state, the model is optimized to predict the oracle's action via maximum-likelihood estimation. In our setting, the state is the coordinator's hidden-state representation of the input, and the action is the index of the selected LLM.

**Dataset.** We first extract the labels from our per-question-best oracle results. Specifically, each label is generated by first identifying, for each seed independently, which LLM achieved the highest reward on that question. When multiple LLMs tie at the maximum reward, we uniformly sample one from the tied set. We then aggregate these per-seed selections across all seeds via majority voting. The LLM selected most frequently across seeds becomes the final label for that question. In cases where multiple LLMs receive equal votes, we uniformly sample from the tied candidates to ensure unbiased label assignment. This approach yields a realistic per-trial performance estimate while maintaining label diversity across the model pool. Table 5 shows the resulting agent label distribution over different tasks.

Table 5: **Agent label distribution by task.** Percentage and count of datapoints where each agent was selected as best via majority vote across seeds.

| Agent | LiveCodeBench | MATH500 | MMLU | RLPR | Overall |
|---|---|---|---|---|---|
| Gemini Pro 2.5 | 17.7% (31) | 18.0% (18) | 16.1% (247) | **17.3%** (898) | **17.1%** (1194) |
| GPT-5 | **39.4%** (69) | 13.0% (13) | **16.4%** (251) | 14.7% (762) | 15.7% (1095) |
| Claude-4-Sonnet | 17.7% (31) | **21.0%** (21) | 14.4% (221) | 14.4% (748) | 14.6% (1021) |
| Qwen3-32B (reasoning) | 7.4% (13) | 12.0% (12) | 12.9% (198) | 15.1% (781) | 14.4% (1004) |
| DeepSeek-R1-Qwen-32B | 3.4% (6) | 15.0% (15) | 14.6% (224) | 14.5% (750) | 14.2% (995) |
| Qwen3-32B (direct) | 10.9% (19) | 17.0% (17) | 16.3% (249) | 14.3% (739) | 14.6% (1024) |
| Gemma-3-27B-IT | 3.4% (6) | 4.0% (4) | 9.2% (141) | 9.8% (506) | 9.4% (657) |

**Training.** We optimize the coordinator using Adam (Kingma & Ba, 2017) with the frozen SLM, training only the linear head. After experimenting with various learning rates and batch sizes, we found that a learning rate of $1 \times 10^{-6}$ and batch size of 64 yield the best coordinator performance. The trained coordinator achieves scores of 0.592, 0.786, 0.906, and 0.360 on LiveCodeBench, MATH500, MMLU, and RLPR respectively. Figure 7 shows the learned agent selection distribution, illustrating which agents the coordinator preferentially select for each task type.

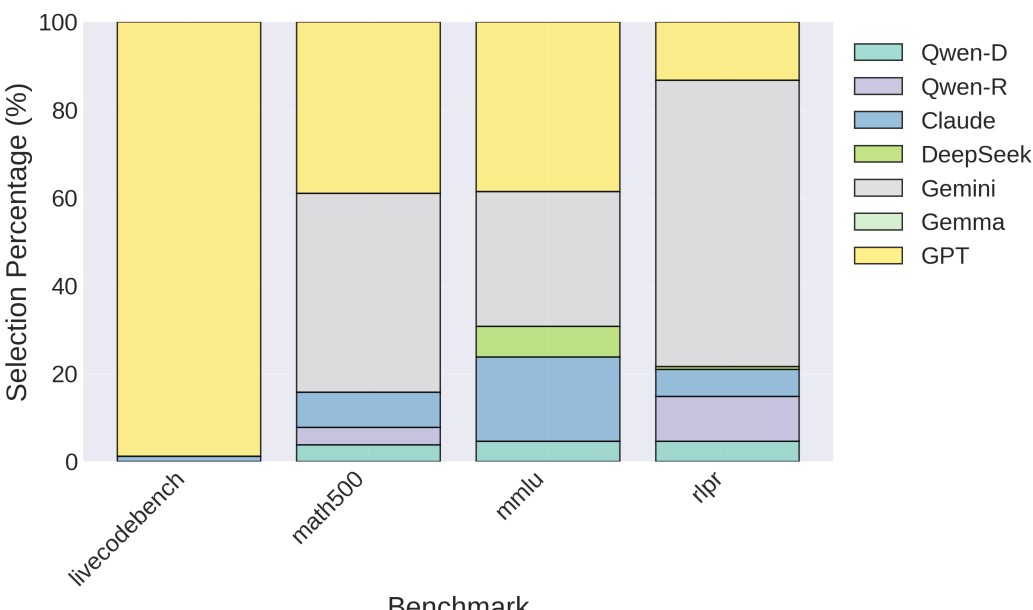

Figure 7: **Agent selection distribution by task.** Percentage of datapoints where each agent was selected by the trained coordinator.

### A.2.2 COST IN LABEL GENERATION

The cost profiles of SFT and label-free training methods, such as sep-CMA-ES, REINFORCE, and RS differ substantially. For SFT, the dominant cost lies in label generation. Labels can be produced at reasonable cost for a direct mapping from representation space to single-step agent selection, but become quickly intractable in multi-turn settings. In our direct-mapping setting, generating labels requires running 3 seeds on 7k datapoints across 7 agents, resulting in $3 \times 7k \times 7 = 147k$ LLM queries.

For multi-turn coordination, the label complexity grows exponentially. Under our experimental configuration with up to 5 turns and 7 candidate agents per turn, the number of required LLM queries for agent selection alone scales by a factor of $7^4 \approx 2.4 \times 10^3$ relative to the single-step setting. Moreover, in multi-turn settings the role selection (among 3 roles) is also relevant at each of the 5 turns, introducing an additional factor of $3^5 = 243 \approx 2.4 \times 10^2$. In total, this yields a multiplicative factor of $7^4 \cdot 3^5 = 583,443 \approx 5.8 \times 10^5$, inflating the cost to an enormous $1.5 \times 10^5 \times 5.8 \times 10^5 \approx$

$8.7 \times 10^{10}$ LLM queries. By contrast, label-free training methods such as sep-CMA-ES require no explicit label generation and instead optimize the coordinator directly based on task rewards.

In summary, while SFT can provide performance gains for a direct representation-to-agent mapping, its prohibitive label-generation cost makes it unsuitable for training multi-turn coordinators, limiting its scalability.

## A.3 FULL ANALYSIS OF SEPARABILITY IN REPRESENTATION SPACE

This section examines how well the extracted hidden states and the coordinator's output logits separate relevant classes. For hidden states, greater separability implies that the SLM's representations encode richer context, providing a stronger signal for the lightweight head to make task-aware decisions.

First, we examine separability along three complementary axes: (i) *Notion of separability*: linear vs. non-linear; (ii) *Label source*: task-type labels (from metadata; input-side) vs. agent/role selection labels (from the head's logits; decision-side); (iii) *Feature space*: raw SLM hidden states (representation space) vs. the coordinator head's output logits (coordination space).

For each cross-combination, we use standard dimensionality-reduction visualizations (PCA/LDA for linear structure; t-SNE/UMAP for non-linear structure) and report classification accuracy using linear and RBF SVMs as quantitative proxies for linear and non-linear separability, respectively. Features are standardized; visualizations are used qualitatively, and SVM accuracies provide the quantitative assessment. Figures 8–13 summarize the results.

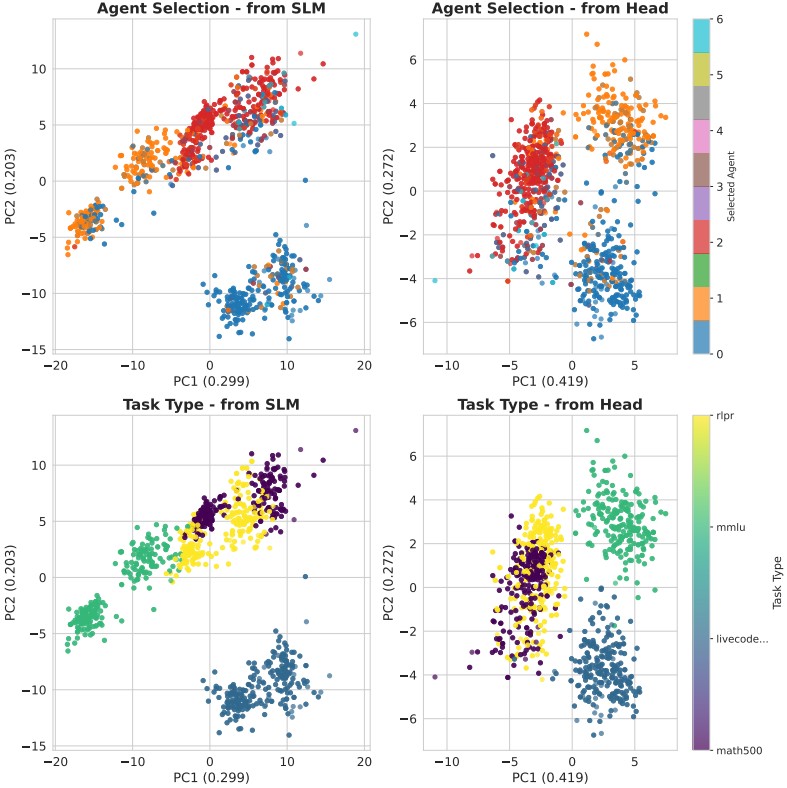

Figure 8: **PCA analysis.** All four plots demonstrate clear clustering patterns.

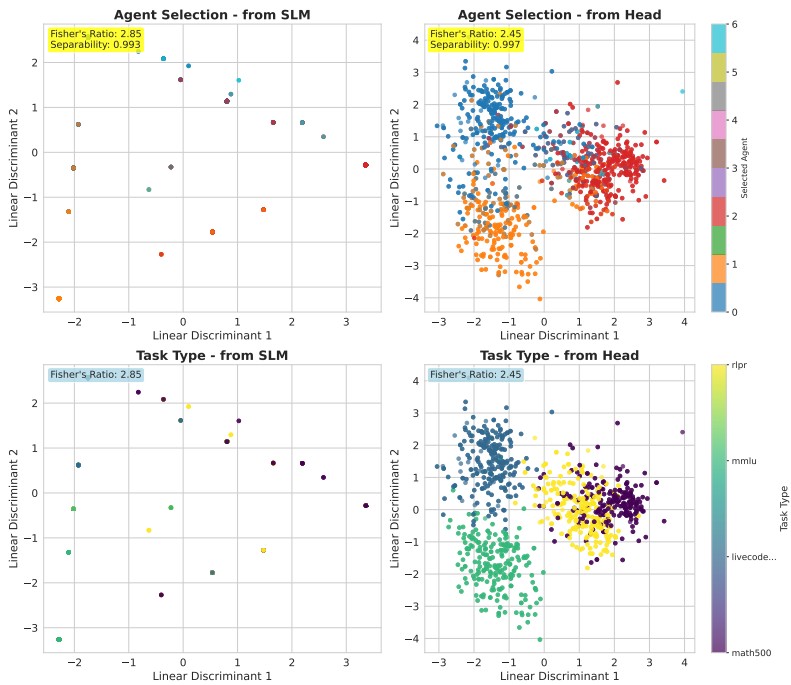

Figure 9: **LDA analysis.** The Fisher's ratios indicate that the between-class scatter is approximately two to three times greater than the within-class scatter.

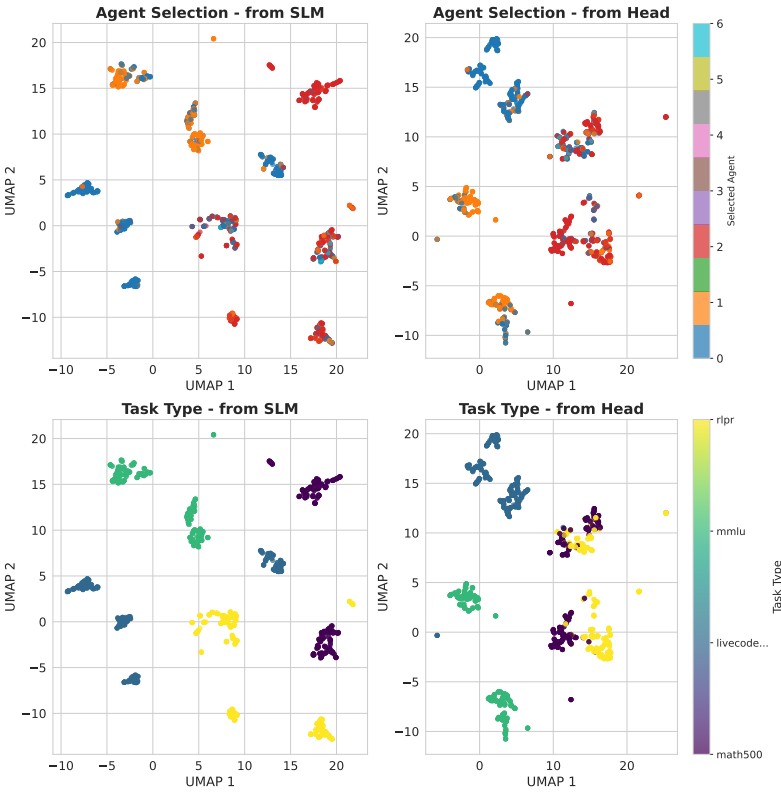

Figure 10: **UMAP analysis.** The clustering patterns indicate strong non-linear separability.

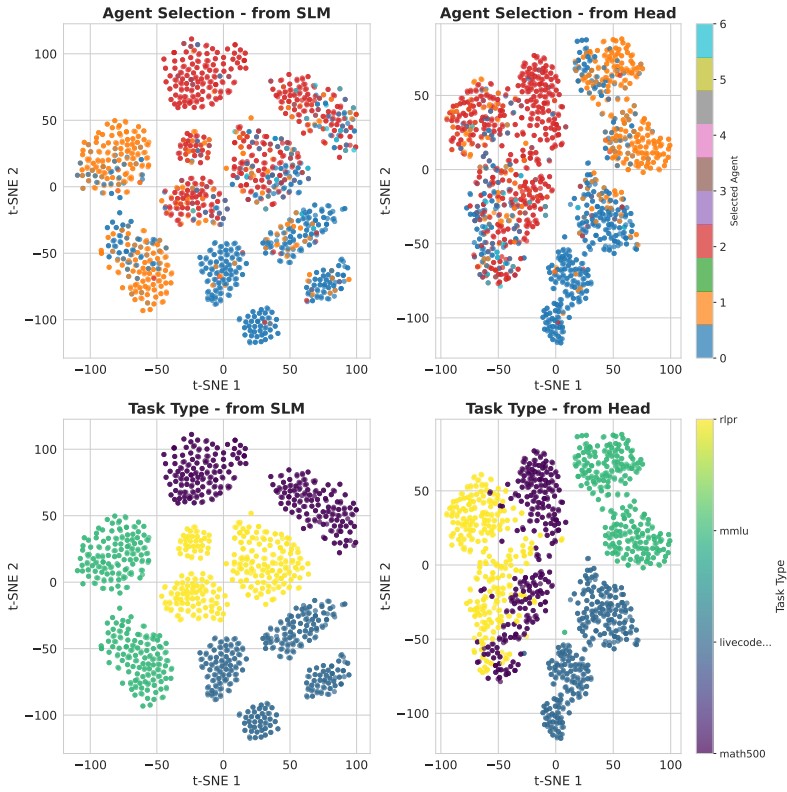

Figure 11: **t-SNE analysis.** The analysis demonstrates particularly strong separability of task types in the hidden states extracted from the SLM.

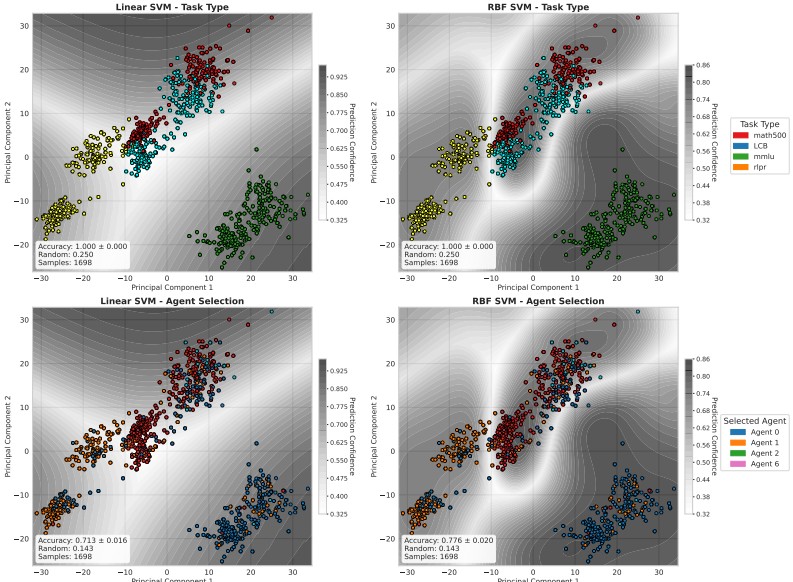

Figure 12: **SVM analysis on hidden states extracted from the SLM.** Classification accuracies: Linear SVM (task type) = 1.000, RBF SVM (task type) = 1.000, Linear SVM (agent selection) = 0.713, RBF SVM (agent selection) = 0.776.

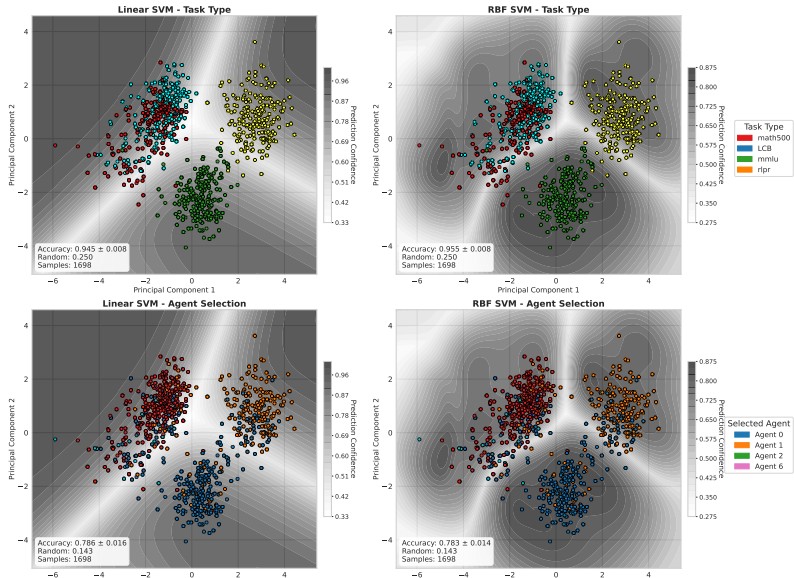

Figure 13: **SVM analysis on output logits.** Classification accuracies: Linear SVM (task type) = 0.945, RBF SVM (task type) = 0.955, Linear SVM (agent selection) = 0.786, RBF SVM (agent selection) = 0.783.

From Figures 8–11, both linear (PCA/LDA) and non-linear (UMAP/t-SNE) views reveal clear structure. LDA's reported Fisher ratios (between/within scatter $\approx$2–3$\times$) corroborate that much of the variance aligns with task-discriminative directions, while PCA shows separation already in the top components, suggesting a substantial linearly aligned subspace.

The SVM results (Figures 12–13) are especially revealing: in the *representation space*, task-type classification is near-perfect for both linear and RBF kernels, implying that penultimate-token hidden states encode task semantics in a nearly linearly separable manifold even after standardization and class-balance controls. In the *coordination space* (head logits), task-type accuracy decreases while agent/role-selection accuracy increases (notably for the linear SVM, which aligns with the head *linear* (see Appendix A.4)), indicating that the head compresses and reorients input semantics toward low-dimensional, decision-aligned axes. This redistribution is consistent with a policy that projects context onto agent-specific logit directions, yielding simpler, more linearly separable boundaries for agent selection.

Next, we investigate how representation space separability relates to coordinator performance. We train linear SVMs to predict agent selections from the hidden states extracted from the SLM , using the coordinator's agent selection as labels. Across the four datasets, LiveCodeBench, MATH500, MMLU, and RLPR, the classification accuracies are 0.844, 0.764, 0.679, and 0.544, respectively.

This ranking aligns with our experimental findings: Sections 4.2 and 4.4 demonstrate that TRINITY shows stronger performance advantages over baseline methods on LiveCodeBench and MATH500 compared to MMLU and RLPR. While these agent selection labels reflect the coordinator's learned behavior rather than ground truth assignments, the correlation between classification accuracy and relative performance gains suggests that tasks exhibiting greater separability in the representation space may be more amenable to effective coordination.

To directly examine the relationship between the intrinsic separability among the datapoints in one task in the representation space and the coordinator's performance, we conduct a controlled experiment using synthetic datasets. Directly controlling separability in real task distributions is impractical, as interventions such as injecting noise into hidden states may introduce confounding factors beyond separability changes (e.g., distributing samples out-of-distribution or altering semantic structure). Therefore, we generate synthetic datasets that replicate the exact structure of the coordinator's representation space (1024 dimensions, 7 agent classes, 4 task type clusters) while systematically varying separability levels. We control separability by systematically scaling the distances between

class centers while maintaining consistent within-class covariance, generating datasets whose measured separability index (between-class variance / total variance) vary.

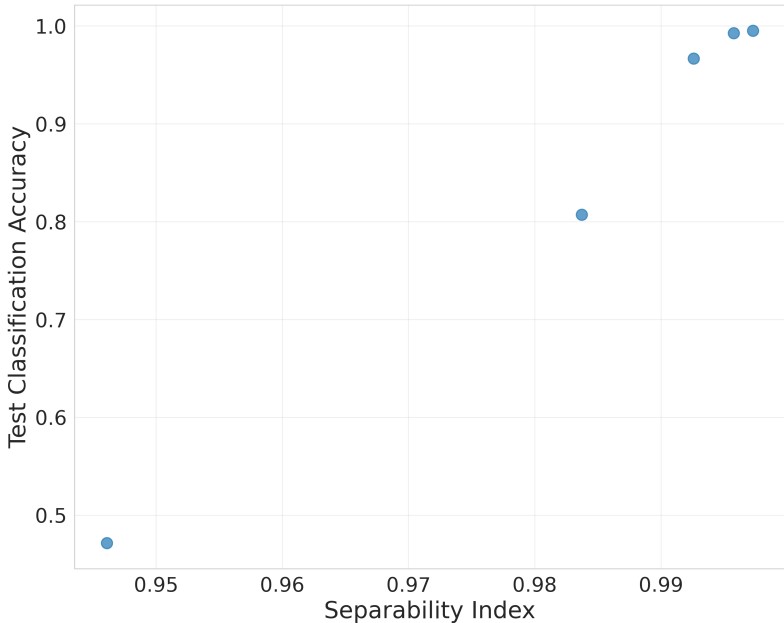

Figure 14: **Separability index vs head classification accuracy.** Trained on synthetic datasets with systematically varied separability, the head *linear* exhibits a strong positive correlation between separability index and test classification accuracy.

We train the exact same head used in our experiments, *linear*, on these synthetic datasets. Figure 14 reveals a strong positive correlation between the separability index and the head's classification accuracy, with test classification accuracy increasing steadily as separability index rises. This controlled experiment indicates that higher intrinsic separability for a task in the representation space enables better head's performance on the task, independent of task-specific confounds.

### A.4   HEAD ARCHITECTURE DESIGN

We describe four heads that maps the SLM's hidden state $\mathbf{h} \in \mathbb{R}^{d_h}$ to agent and role selection logits $\mathbf{z} \in \mathbb{R}^{n_a}$, subsequently turned into probabilities with a softmax or argmax.

*Linear* head refers to the most direct affine mapping (without bias) from hidden states to logits. It computes

$$\mathbf{z} = \mathbf{W}\mathbf{h}, \qquad \mathbf{W} \in \mathbb{R}^{n_a \times d_h}. \tag{5}$$

This head has exactly $d_h n_a$ trainable parameters and serves as a strong baseline. It allows unrestricted linear combinations of hidden dimensions to express agent and role preferences while remaining simple and fast to train.

*Low-rank* head refers to a factorized bottleneck with a nonlinearity that replaces a single dense map by two smaller projections. We use

$$\mathbf{u} = \text{ELU}(\mathbf{U}\mathbf{h}), \qquad \text{ELU}(x) = \begin{cases} x, & x \geq 0 \\ \alpha(e^x - 1), & x < 0 \end{cases}, \ \alpha = 0.1, \tag{6}$$

$$\mathbf{z} = \mathbf{V}\mathbf{u} \cdot \sigma, \tag{7}$$

with $\mathbf{U} \in \mathbb{R}^{r \times d_h}$, $\mathbf{V} \in \mathbb{R}^{n_a \times r}$, and a fixed non-trainable scale $\sigma \in \mathbb{R}$. In this work we *fix* the bottleneck to $r = 14$. This choice can result in *more* parameters than a strictly compressed low-rank setting, but it intentionally adds depth and nonlinearity so the head can capture non-linear patterns at

reduced per-projection cost versus a single wide mapping. We initialize with Xavier-uniform (Glorot & Bengio, 2010) using adaptive gains:

$$\mathbf{U} \sim \mathcal{U}\left(-\sqrt{\tfrac{6}{d_h+r}}, \sqrt{\tfrac{6}{d_h+r}}\right), \qquad \mathbf{V} \sim \mathcal{U}\left(-\sqrt{\tfrac{18}{r+n_a}}, \sqrt{\tfrac{18}{r+n_a}}\right). \tag{8}$$

*Sparse* head refers to a learnable dimension-selection mechanism that gates hidden features before a linear projection. The logits are

$$\mathbf{z} = \mathbf{W}\left(\mathbf{h} \odot \boldsymbol{\alpha}\right), \qquad \mathbf{W} \in \mathbb{R}^{n_a \times d_h}, \tag{9}$$

where $\boldsymbol{\alpha} \in \mathbb{R}^{d_h}$ is a data-agnostic, learnable selection vector. The target number of active dimensions is $k = \max\big(1, \lfloor d_h \cdot (1 - \mathrm{sigmoid}(\rho)) \rfloor\big)$ with a learnable sparsity logit $\rho$. During training we form a differentiable top-$k$ mask by sampling Gumbel noise and sharpening with a temperature $\tau \in [1.0,\ 20.0]$:

$$\tilde{\mathbf{s}} = (\mathbf{s} + \boldsymbol{\epsilon})/\tau, \quad \boldsymbol{\epsilon} \sim \mathrm{Gumbel}(0,1), \tag{10}$$

$$\boldsymbol{\alpha}_{\mathrm{soft}} = \mathrm{TopK}_{\mathrm{soft}}(\tilde{\mathbf{s}}, k), \qquad \boldsymbol{\alpha} = \frac{\boldsymbol{\alpha}_{\mathrm{soft}} \cdot k}{\sum_{i=1}^{d_h} \alpha_{\mathrm{soft},i}}. \tag{11}$$

At inference, we use a hard top-$k$ binary mask $\boldsymbol{\alpha} = \mathrm{TopK}_{\mathrm{hard}}(\mathbf{s}, k)$. This head has $d_h n_a + d_h + 2$ parameters (projection weights, importance scores, temperature, and sparsity logit) and offers both regularization and interpretability by exposing which hidden dimensions drive agent and role selections.

*Block-diagonal* head refers to structuring the projection matrix with disjoint blocks that couple only subsets of hidden dimensions to subsets of agents or roles:

$$\mathbf{W} = \begin{bmatrix} \mathbf{W}_1 & \mathbf{0} & \cdots & \mathbf{0} \\ \mathbf{0} & \mathbf{W}_2 & \cdots & \mathbf{0} \\ \vdots & \vdots & \ddots & \vdots \\ \mathbf{0} & \mathbf{0} & \cdots & \mathbf{W}_B \end{bmatrix}, \quad \mathbf{z} = \begin{bmatrix} \mathbf{W}_1 \mathbf{h}_1 \\ \mathbf{W}_2 \mathbf{h}_2 \\ \vdots \\ \mathbf{W}_B \mathbf{h}_B \end{bmatrix}, \tag{12}$$

with $\mathbf{h} = [\mathbf{h}_1; \ldots; \mathbf{h}_B]$, $\mathbf{W}_i \in \mathbb{R}^{a_i \times h_i}$. We use two concrete variants. *Block-diagonal-2* sets $B = 2$ and partitions both hidden and agent/role dimensions proportionally, e.g.,

$$a_i = \min\left(\left\lceil \tfrac{n_a}{2} \right\rceil, n_a - \sum_{j<i} a_j\right), \qquad h_i = \begin{cases} \lfloor \tfrac{a_i d_h}{n_a} \rfloor, & i < 2 \\ d_h - \sum_{j<2} h_j, & i = 2 \end{cases}.$$

*Block-diagonal-10* denotes the high-independence case corresponding to our setting with $n_a = 10$ logits. It creates one block per agent/role ($B = 10$, $a_i = 1$) and distributes hidden dimensions as evenly as possible, yielding

$$z_j = \mathbf{w}_j^\top \mathbf{h}_j, \qquad h_j = \begin{cases} \lfloor \tfrac{d_h}{10} \rfloor + 1, & j \le (d_h \bmod 10) \\ \lfloor \tfrac{d_h}{10} \rfloor, & \text{otherwise} \end{cases}.$$

*Block-diagonal-2* blocks moderate amount of parameter correlations, whereas *block-diagonal-10* maximizes independence across the ten logits.

Table 6: **Parameter size distribution in training.** The size is calculated based on the SLM Qwen3-0.6B. SVF refers to singular value fine-tuning.

| | SVF | *linear* | *low-rank* | *sparse* | *block-diagonal-2* | *block-diagonal-10* |
|---|---|---|---|---|---|---|
| **Parameter Size** | 9216 | 10240 | 20680 | 11266 | 5120 | 1024 |

Table 6 compares the parameter counts of the different head architectures alongside the parameters trained in singular value fine-tuning. *Block-diagonal-10* achieves an exact $10\times$ reduction in head parameters relative to *linear* (1,024 vs. 10,240 parameters for $d_h = 1024$, $n_a = 10$). In contrast, *low-rank* replaces the single $d_h \times n_a$ projection with two matrices $\mathbf{U} \in \mathbb{R}^{r \times d_h}$ and $\mathbf{V} \in \mathbb{R}^{n_a \times r}$ (with $r = 14$) and an ELU nonlinearity, increasing the head size to 20,680 parameters. This is roughly a $2\times$ increase over *linear*, trading parameter efficiency for additional depth and non-linearity in the mapping from hidden states to logits.

## A.5 EXPERIMENTATION WITH LEARNING ALGORITHMS

We also compare our learning strategy with the REINFORCE algorithm and RS with fitness averaging. To ensure the total evaluation budgets were equivalent, we configured the baselines as follows. For REINFORCE, we used a batch size equal to the per-iteration evaluation size of sep-CMA-ES and ran for 60 iterations. For RS, we performed 32 trials for each sampled parameter vector, continuing until the total number of trials matched the evaluation count of sep-CMA-ES.

For RS, we warmstart it by calibrating the sampling range using the high-performing weights obtained via sep-CMA-ES. Specifically, we sample uniformly from $[-0.5, 0.5]$, a band that slightly exceeds the observed extrema of those weights. For each sampled parameter vector, we run 32 independent trials and compare the average reward.

## A.6 DATASET–AGENT SUBSET SELECTION

To construct a pool of complementary agents and a curriculum of datasets that together amplify coordination gains, we cast selection as a joint subset selection over datasets and agents. Our formulation and procedure adhere to two principles: (i) evaluate gains in the error space to capture practical improvements across varying accuracy regimes; (ii) enforce complementarity, not merely strength, so the coordinator can exploit heterogeneous capabilities.

### OBJECTIVE: MAXIMIZE RELATIVE ERROR REDUCTION UNDER JOINT CONSTRAINTS

Let $\mathcal{M} = \{M_1, \ldots, M_X\}$ be candidate agents and $\mathcal{D} = \{D_1, \ldots, D_Y\}$ be candidate datasets. Let $E(D_y, M_x) \in [0, 1]$ denote the observed accuracy of agent $M_x$ on dataset $D_y$ under a fixed inference protocol (without coordination, identical output-token budget, and prompting). For any dataset subset $C \subseteq \mathcal{D}$ and agent subset $\mathcal{M}' \subseteq \mathcal{M}$, define

$$Z_{C,\mathcal{M}'} = \frac{1}{|C|} \sum_{D_y \in C} \max_{M_x \in \mathcal{M}'} E(D_y, M_x), \qquad S^*_{C,\mathcal{M}'} = \max_{M_x \in \mathcal{M}'} \frac{1}{|C|} \sum_{D_y \in C} E(D_y, M_x). \quad (13)$$

Here, $Z_{C,\mathcal{M}'}$ denotes the *combination* performance—i.e., the best-per-dataset accuracy obtained by coordinating each $D_y \in C$ to its highest-performing agent in $\mathcal{M}'$—and thus ignores potential synergistic interactions among agents within a dataset. While $Z_{C,\mathcal{M}'}$ may not fully reflect end-to-end coordinated performance, it serves as a tractable proxy that is typically positively correlated with it. In contrast, $S^*_{C,\mathcal{M}'}$ is the *best single-agent* baseline on the same $C$, obtained by fixing one agent in $\mathcal{M}'$ for all datasets. We then optimize the *relative error reduction* (RER):

$$\text{RER}(C, \mathcal{M}') = \frac{Z_{C,\mathcal{M}'} - S^*_{C,\mathcal{M}'}}{1 - S^*_{C,\mathcal{M}'}}. \quad (14)$$

This criterion rewards settings where no single agent dominates across all datasets and where specialization materially lowers error. Our joint selection problem is

$$(C^*, \mathcal{M}^*) \in \arg \max_{C \subseteq \mathcal{D},\, \mathcal{M}' \subseteq \mathcal{M}} \text{RER}(C, \mathcal{M}'). \quad (15)$$

### JOINT DATASET–AGENT SUBSET SELECTION

For each dataset $D_y$ and a chosen model subset $\mathcal{M}' \subseteq \mathcal{M}$, the *best model for the individual dataset* (doubly constrained) is

$$M^*_{y,\mathcal{M}'} = \arg \max_{M_x \in \mathcal{M}'} E(D_y, M_x). \quad (20)$$

Given subsets $C \subseteq \mathcal{D}$ with $|C| \leq Y$ and $\mathcal{M}' \subseteq \mathcal{M}$ with $|\mathcal{M}'| \leq X$, the *joint combination strategy performance* averages the per-dataset best-in-subset performance:

$$Z_{C,\mathcal{M}'} = \frac{1}{|C|} \sum_{D_y \in C} E(D_y, M^*_{y,\mathcal{M}'}). \quad (21)$$

In contrast, the *single-model performance on the dataset combination* fixes one model $M_x \in \mathcal{M}'$ for all datasets in $C$:

$$S_{x,C} = \frac{1}{|C|} \sum_{D_y \in C} E(D_y, M_x). \quad (22)$$

The *best single model for the joint combination* is therefore

$$M^*_{C,\mathcal{M}'} = \arg \max_{M_x \in \mathcal{M}'} S_{x,C},$$
(23)

with corresponding performance

$$S^*_{C,\mathcal{M}'} = S_{M^*_{C,\mathcal{M}'},C} = \max_{M_x \in \mathcal{M}'} S_{x,C}.$$
(24)

**Problem.** Find the optimal subsets $(C^*, \mathcal{M}^*)$ that maximize the relative error reduction:

$$(C^*, \mathcal{M}^*) = \arg \max_{\substack{C \subseteq \mathcal{D}, |C| \leq Y \\ \mathcal{M}' \subseteq \mathcal{M}, |\mathcal{M}'| \leq X}} \frac{Z_{C,\mathcal{M}'} - S^*_{C,\mathcal{M}'}}{1 - S^*_{C,\mathcal{M}'}}$$
(25)

**Equivalently:**

$$(C^*, \mathcal{M}^*) = \arg \max_{\substack{C \subseteq \mathcal{D}, |C| \leq Y \\ \mathcal{M}' \subseteq \mathcal{M}, |\mathcal{M}'| \leq X}} \frac{(1 - S^*_{C,\mathcal{M}'}) - (1 - Z_{C,\mathcal{M}'})}{1 - S^*_{C,\mathcal{M}'}}$$
(26)

**Expanded form.**

$$\max_{\substack{C \subseteq \mathcal{D} \\ \mathcal{M}' \subseteq \mathcal{M}}} \frac{\frac{1}{|C|} \sum_{D_y \in C} \max_{M_x \in \mathcal{M}'} E(D_y, M_x) - \max_{M_x \in \mathcal{M}'} \frac{1}{|C|} \sum_{D_y \in C} E(D_y, M_x)}{1 - \max_{M_x \in \mathcal{M}'} \frac{1}{|C|} \sum_{D_y \in C} E(D_y, M_x)}$$
(27)

CANDIDATE FILTERING VIA A TOP-5% PERFORMANCE FRONTIER

The joint search space for the problem is combinatorial. We begin by computing the performance matrix, estimating $E(D_y, M_x)$ for all pairs $(D_y, M_x)$ under the standardized protocol. Next, we perform a quantile filter at the top 5%: let $\tau$ denote the 95th percentile of $\{E(D_y, M_x)\}$ across all pairs and define

$$\mathcal{K}_{95} = \{(D_y, M_x) : E(D_y, M_x) \geq \tau\}.$$
(16)

This top-5% filtering concentrates the subsequent selection on strong, demonstrably effective pairings while preserving diversity across tasks.

JOINT SELECTION VIA EXHAUSTIVE ENUMERATION (AND WHEN HEURISTICS ARE NEEDED)

Although joint subset selection over datasets and agents is exponential in general, our experimental regime admitted an *exact* solution. The candidate sets were sufficiently small to permit *exhaustive enumeration* under our evaluation budget. Concretely, we enumerate all pairs $(C, \mathcal{M}')$ satisfying the coverage constraint, compute $Z_{C,\mathcal{M}'}$, $S^*_{C,\mathcal{M}'}$, and $\text{RER}(C, \mathcal{M}')$ for each, and select $(C^*, \mathcal{M}^*)$ maximizing RER. When two candidates exhibit statistically indistinguishable RER, we break ties by prioritizing diversity in task and agent types. For example, we favor a balanced mixture of reasoning agents and agents with direct inference capabilities.

Exhaustive enumeration scales poorly as $|\mathcal{D}| + |\mathcal{M}|$ grows; beyond moderate sizes, even after frontier pruning, the search can become prohibitive. In such regimes, the same objective can be pursued with budget-aware heuristics (e.g., greedy seeding followed by annealed or beam-style refinement) while retaining the coverage constraint and the complementarity-based tie-breaking.

## A.7 EXPERIMENTAL DETAILS.

### A.7.1 BASELINE SETUP

- **Individual Agent:** We compare against the strongest individual models in our agent pool—GPT-5, Gemini-2.5-pro, and Claude-4-Sonnet—evaluated at both 4K and 20K(5x) maximum token limits to account for the accumulated context at each hop in our multi-turn framework.
- **Random Agent Selection:** A simple baseline where an agent is selected randomly at each turn during the multi-turn collaboration process, providing a lower bound for structured agent coordination. And the max turn number is 5, same as TRINITY setting.

- **Self-Reflection:** An extended version of standard reflection where a single agent produces an initial answer and then reflects on its own output over five turns, representing iterative self-improvement without collaboration with others.

- **MasRouter:** A recently method trained using the same dataset as our approach, with model selection based on best validation loss. The training follows recommend settings and employs cost-regularization and as detailed in the original paper, using the MMRL dataset with 256 samples, validating every 5 epochs, and selecting the best checkpoint after observing sufficient evidence of overfitting.

- **RouterDC:** A routing method trained on 500 samples from the MMRL dataset to match the conditions specified in the original paper. Each sample is repeated 5 times to collect average performance across all workers for a given question, with this average performance incorporated as part of the training label.

- **Smoothie:** Applied as a test-time method to questions and outputs from each agents, evaluated under both dependent strategies (selecting one agent per individual question) and independent strategies (selecting one single agent for the entire test set).

- **Mixture of Agents (MoA):** Implemented as a test-time scaffold with a single MoA layer and single aggregator layer, totaling 8 model calls per question, where a random model is chosen to serve as the final aggregator.

- **Per Question Best:** A theoretical upper bound representing the optimal performance achievable by correctly selecting the best-performing worker model for each individual question, providing the argmax baseline for comparison.

### A.7.2 AGENT DISTRIBUTION ACROSS TASKS.

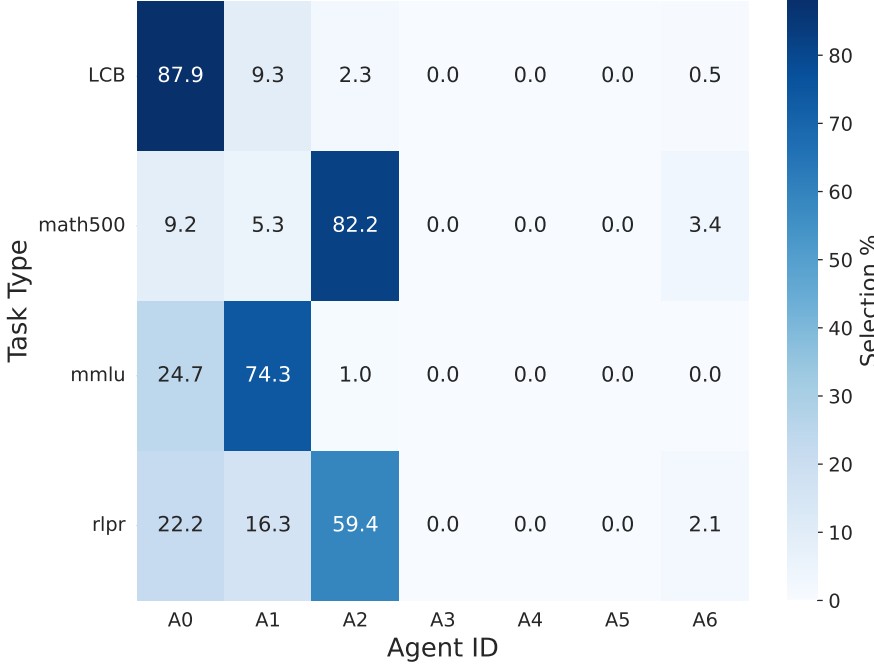

Figure 15: **Agent distribution over tasks.** A0: GPT-5, A1: Claude-Sonnet-4-20250514, A2: Gemini-2.5-pro, A3: DeepSeek-R1-Distill-Qwen-32B, A4: Gemma-3-27b-It, A5: Qwen3-32B (reasoning), A6: Qwen/Qwen3-32B (direct). TRINITY demonstrates strong task-aware agent selection strategy.

### A.7.3 ADDITIONAL BASELINE RESULTS.

**Parallel Sampling.** We report additional baselines using majority voting over 5 samples per question. While TRINITY is designed to handle a broad spectrum of tasks, majority voting with 5 samples

is only applicable to settings with a small, discrete set of candidate outputs, such as multiple-choice benchmarks. Table 7 summarizes the resulting performance on MMLU.

Table 7: **Majority@5 baseline results for MMLU.** For each question, the answer is chosen based on a majority voting over 5 parallel inquiries.

| Model | Avg Score |
|---|---|
| Gemini Pro 2.5 | **91.57** $\pm$ 0.70 |
| GPT-5 | 91.31 $\pm$ 0.23 |
| Claude-4-Sonnet | 90.99 $\pm$ 0.39 |

**LLM as Coordinator.** We also evaluated an approach where an LLM is directly prompted to select the model and role at each turn. Gemini Pro 2.5 was chosen as the coordinator given its superior performance among agents. However, this prompting-based method underperforms TRINITY's trained coordinator (64.14 vs 70.44 average score). We observe that the LLM struggles to comprehend and manage the properties of all 7 agents, resulting in inconsistent and suboptimal selections. This demonstrates that prompting with closed sourced LLMs is insufficient for capturing agents' inherent characteristics, which a coordinator acquires through training. See Table 8 for detailed results.

Table 8: **Comparison between TRINITY and LLM as Coordinator**

| Method | Math500 | MMLU | RLPR | LiveCodeBench | Avg |
|---|---|---|---|---|---|
| TRINITY | 88.00 | 91.56 | 40.72 | 61.49 | 70.44 |
| Gemini 2.5 pro as Coordinator | 78.67 | 83.26 | 26.83 | 26.28 | 53.76 |

### A.7.4 TOKEN USAGE TABLES ON IN-DISTRIBUTION TASKS.

Table 9: **Average output token number of coordination methods**

| Model | Math500 | MMLU | RLPR | LiveCodeBench |
|---|---|---|---|---|
| TRINITY | 2,853 | 1,200 | 2,141 | 1,999 |
| MOA | 6,871 | 5,218 | 11,086 | 21,634 |
| RouterDC | 624 | 374 | 811 | 1,552 |
| Smoothie | 6,472 | 4,718 | 10,580 | 17,864 |
| MASRouter | 4,260 | 1,847 | 5,370 | 8,401 |

Table 10: **Average output token number of each model in 5× Self-Reflection**

| Model | Math500 | MMLU | RLPR | LiveCodeBench |
|---|---|---|---|---|
| Qwen3-32B (direct) | 2,075 | 1,746 | 1,949 | 4,207 |
| Qwen3-32B (reasoning) | 2,692 | 2,213 | 3,349 | 7,575 |
| DeepSeek-R1-Distill-32B | 3,988 | 3,811 | 4,609 | 12,228 |
| Gemma-3-27B | 1,704 | 820 | 1,750 | 3,714 |
| Claude Sonnet 4 | 1,834 | 1,293 | 1,580 | 3,210 |
| GPT-5 | 577 | 428 | 895 | 1,971 |
| Gemini-2.5-Pro | 5,142 | 5,460 | 6,710 | 11,046 |

Table 11: **Average output token number of each model in 5× Context**

| Model | Math500 | MMLU | RLPR | LiveCodeBench |
|---|---|---|---|---|
| Qwen3-32B (direct) | 447 | 152 | 392 | 192 |
| Qwen3-32B (reasoning) | 1,019 | 382 | 1,047 | 1,784 |
| DeepSeek-R1-Distill-32B | 1,343 | 538 | 1,369 | 4,066 |
| Gemma-3-27B | 342 | 146 | 336 | 159 |
| Claude Sonnet 4 | 367 | 218 | 300 | 518 |
| GPT-5 | 221 | 66 | 219 | 1,207 |
| Gemini-2.5-Pro | 1,153 | 579 | 787 | 5,753 |

Table 12: **Average output token number of each model in Default Context (4096)**

| Model | Math500 | MMLU | RLPR | LiveCodeBench |
|---|---|---|---|---|
| Qwen3-32B (direct) | 521 | 154 | 406 | 419 |
| Qwen3-32B (reasoning) | 995 | 397 | 1,191 | 1,789 |
| DeepSeek-R1-Distill-32B | 1,175 | 485 | 1,181 | 3,443 |
| Gemma-3-27B | 437 | 147 | 330 | 483 |
| Claude Sonnet 4 | 382 | 217 | 304 | 530 |
| GPT-5 | 218 | 66 | 220 | 1,113 |
| Gemini-2.5-Pro | 819 | 578 | 774 | 2,396 |

