# OpenReview forum: "Trinity: An Evolved LLM Coordinator"
_ICLR.cc/2026/Conference — ICLR 2026 Poster_

### Official Review · Reviewer_bpif · 2025-10-28

**Soundness:** 3
**Presentation:** 2
**Contribution:** 3
**Rating:** 6
**Confidence:** 4

**Summary:**

This paper proposes a three-stage framework by calling LLMs to collaboratively solve a problem. Their method, Trinity, interacts with several LLMs by assigning one of three roles (thinker, worker, and verifier). They demonstrate consistent improvements over individual frontier LLMs on a variety of math, code, and reasoning benchmarks.

**Strengths:**

* The paper achieves good performance on benchmarks (to the best of my knowledge, 86.2% is indeed state of the art on LiveCodeBench).
* To the best of my knowledge, this is the first work I have seen that applies LLM routing to a multi-turn setting, so I think this approach is novel.
* The study is comprehensive, with ablation studies for different design choices, and interpretability study to demonstrate the effectiveness of delgating tasks to different LLMs.

**Weaknesses:**

I am concerned about the organization of the paper. Some notes:
* Section 2 (problem formulation) feels very short. Some notation is not properly introduced: what is the definition of $B_{turn}$? how is it related to turns $T$? what is $B_{env}$, and what does it mean for the budget to be atomic?
* Figure 2 could have better presentation - it is not immediately obvious that the penultimate output token is a part of the input text (which is my understanding)
* Section 3.2: too many important details are moved to the appendix. For example,when you move the empirical analysis to the appendix, it seems like it is more of an afterthought rather than properly trying to motivate the use of evolutionary strategies.
* Figure 3 is hard to follow, not enough visual prominence to show your method over others

The choice of singular-value finetuning for the coordinator head seems a bit strange. To my understanding, the method is using so as to make the number of learnable parameters extremely small. Yet in appendix A.3.2 you show that linear fine-tuning (e.g. just applying a full-rank matrix head) has the best performance. I am a little bit confused by the story here; in my mind, given the coordinator model is already quite small (0.6B), I would rather have full-power finetuning on the head to maximize performance than to save a few learnable parameters.

**Questions:**

* I would like to have further discussion on the ablation study for removing the tri-role selection. I assume that you mean instead of assigning three roles, you directly route the query to one LLM and have it solve the problem (so essentially only 1 role). My question is what happens if you have only two (e.g. merge the thinker and worker together, and one verifier)? In Figure 1 it looks like the worker is simply following the instructions of the thinker, so couldn't we merge the two and get similar results (and also reduce the number of calls you need to make)?

* I am still unsure on why the SLM needs to be trained via RL (again, why I think section 2 is too short). I am sure it works, but why wouldn't it be sufficient to train on simple (state, action) pairs, where the action is just the choice of picking one LLM over another?

---

> ### Author Response · Authors · 2025-11-21
> **Response to Reviewer bpif part 1**
>
> We thank the reviewer for the positive comments on our good performance, methodological novelty, and comprehensive studies. We are also grateful for the constructive feedback to help us further improve the paper. We address the reviewer's concerns in the following. Please do not hesitate to let us know if you have any question.
>
> ---
>
> > **W1:** Concern about the organization of the paper.
>
> We highly appreciate your careful review and detailed suggestions for organization improvements, which have been incorporated into the revision, we present a summary of the changes below.
>
> - Section 2: Notations are further clarified. $T$ is the realized trajectory horizon (number of turns actually taken), while $B_\mathrm{turn}$ is a fixed maximum turn budget; they are related by $T \le B_\mathrm{turn}$, since trajectories may terminate early when the verifier accepts. $B_\mathrm{env}$ is a computational budget constraining total evaluations. An "atomic evaluation" refers to one complete end-to-end trajectory sample (yielding a single Bernoulli reward), which internally requires multiple LLM calls; the atomic budget $B_\mathrm{env}$ counts these complete evaluations.
> - Figure 2: The penultimate token is highlighted in blue in the figure as head input, and related details have also been added to the figure caption.
> - Section 3.2: We agree on the importance of this empirical analysis. It has been restored to the main text as Section 4.7 (previously moved to the appendix due to length constraints). Also, we improved the description about our thought process for sep-CMA-ES in Section 3.3.
> - Figure 3: The text label of our method has been updated with bold text and a larger font size.
>
> ---
>
> > **W2:** The choice of singular-value finetuning for the coordinator head seems a bit strange.
>
> We have revised the writing to clear this ambiguity. As a summary of our revision, we actually used full fine-tuning on the linear head. The singular value fine-tuning (SVF) is applied only to update parameters inside the 0.6B coordinator backbone (specifically, the second-to-last layer of the 0.6B SLM). The head itself is a full-rank linear projection and all of its parameters are updated.
>
> Also, as a side point, contrary to what the head names might suggest, the number of parameters in the *linear* head is actually smaller than in *low-rank*, because the low-rank variant introduces an extra hidden layer (two projection matrices plus a non-linearity) to capture additional non-linearity, whereas the “full-rank” *linear* head is a single projection layer.
>
> We have updated Section 4.1 (Experimental setup) and Appendix A.3.2 in our revision to better clarify this training setting. To help understand how many parameters are being trained where, the table below summarizes the parameter counts for SVF and the different head architectures (all numbers computed for Qwen3-0.6B). The rows in boldface are the settings in our experiments, the rest rows are other possible head choices.
>
> | Component  | \#Parameters |
> | ----------------- | -------------- |
> | **SLM** (SVF training)               | 9,216          |
> | **linear** (full-rank training)       | 10,240         |
> | low-rank          | 20,680         |
> | sparse            | 11,266         |
> | block-diagonal-2  | 5,120          |
> | block-diagonal-10 | 1,024          |

---

> ### Author Response · Authors · 2025-11-21
> **Response to Reviewer bpif part 2**
>
> > **Q1:** What happens if you have only two (e.g. merge the thinker and worker together, and one verifier)?
>
> Following the reviewer's instruction, we have conducted the two-role experiment, merging the thinker and worker into a unified solver paired with a verifier.
>
> As we reduce role complexity, the number of calls indeed decreases. However, the result table below shows this simplification also causes clear average performance degradation as more role components are removed.
>
> | Method | Math500 | MMLU | RLPR | LiveCodeBench| Avg |
> |--------|---------|------|--------|---------------|------|
> | TRINITY | 88.00 | 91.56 | 40.72 | 61.46 | 70.44 |
> | w/o Thinker-role selection | 86.20 | 92.75 | 38.00 | 57.80 | 68.69 |
> | w/o Tri-role selection | 82.00 | 91.64 | 36.15 | 58.28 | 67.02 |
>
> Notably, on LiveCodeBench, removing all three roles performs similarly to removing just the independent thinker role. This indicates the significance of the thinker role: it enables focused solution planning that produces better plans than a merged solver. Meanwhile, the dedicated worker can concentrate on high-quality execution.
>
> ---
>
> > **Q2:** Why wouldn't it be sufficient to train on simple (state, action) pairs, where the action is just the choice of picking one LLM over another?
>
> Thank you for raising this point. We agree that, in principle, one could train the coordinator on simple (state, action) pairs where the action is the choice of LLM. We now explicitly evaluate this baseline via supervised fine-tuning in a dedicated new section in the Appendix (Appendix A.2).
>
> In the single-step setting, we construct per-question-best oracle labels by running all agents and selecting the highest-reward one (with appropriate tie-breaking), and then train the coordinator to imitate this oracle. As reported in the new SFT section, this supervised baseline is competitive and clearly non-trivial: it outperforms REINFORCE and random search, but still consistently underperforms sep-CMA-ES, even when given access to strong oracle labels. For convienience, this table here compares the performance among different training methods:
>
> | Method      | LiveCodeBench | MATH500 | MMLU  | RLPR  |
> |------------|---------------|---------|-------|-------|
> | REINFORCE  | 0.253         | 0.459   | 0.500 | 0.266 |
> | RS         | 0.374         | 0.794   | 0.897 | 0.345 |
> | SFT        | 0.592         | 0.786   | 0.906 | 0.360 |
> | **sep-CMA-ES** | **0.615** | **0.880** | **0.916** | **0.401** |
>
> The main reason we rely on RL-style feedback is scalability and deployability rather than raw performance. SFT on (state, action) pairs requires explicit oracle actions, which in our setting means running all candidate LLMs (and, in multi-turn settings, all agent/role choices across turns) to determine the best one. Even in the direct single-step mapping, creating these labels already costs $3 \times 7000 \times 7 = 147,000$ LLM queries under our experimental setup. Extending this to multi-turn coordination increases exponetially the LLM queries just for label generation. In contrast, sep-CMA-ES and REINFORCE treat the coordinator as a policy that is updated directly from scalar task rewards along the trajectories it actually induces, without ever needing oracle labels or exhaustive evaluation over all agents and roles. This makes RL-style feedbacks much more scalable to the multi-turn coordination setting in our study.

---

### Official Review · Reviewer_Ev8j · 2025-10-30

**Soundness:** 3
**Presentation:** 3
**Contribution:** 2
**Rating:** 6
**Confidence:** 3

**Summary:**

This paper introduces Trinity, a lightweight coordinator that selects appropriate LLMs and assigns specific roles for the selected LLM to solve the given query in multiple turns. The coordinator, which comprises a 0.6B SLM and a lightweight head, is optimized with sep-CMA-ES, a high efficient strategy for this problem. Experiments indicate that Trinity improves the overall performance on both in-distribution and out-of-distribution benchmarks in several tasks, and the sep-CMA-ES strategy significantly outperforms RL and RS for the optimization of the coordinator.

**Strengths:**

1. The Trinity framework is lightweight and concise, which facilitates the cooperation of multiple state-of-the-art LLMs to solve user queries. It also has a good generalization on several tasks including math, coding, reasoning and domain knowledge, indicating the potential of application on different tasks.
2. The sep-CMA-ES strategy significantly outperforms the RL and RS methods for optimizing the coordinator, demonstrating its high efficiency.

**Weaknesses:**

1. The effectiveness of the Trinity framework is limited compared to single-model baselines. As shown in Figure 3 and Table 1, compared with Gemini Pro 2.5, Trinity only brings an improvement about 1%-3% on all the in-distribution and out-of-distribution benchmarks except LiveCodeBench (where it outperforms GPT-5 for merely 3%). Considering the cost of training a coordinator and multi-turn interaction for selecting models, it shows limited effectiveness compared with simply selecting LLMs according to their adept tasks (e.g. select GPT-5 for coding tasks while Gemini Pro 2.5 for others).
2.  The necessity of LLM selection is not demonstrated in the ablation study. It is unclear whether the performance will degrade when the Tri-role selection process in the Trinity framework is applied on a fixed single-model baseline instead of a selected LLM.

**Questions:**

1. In Section 4.6, how does the separability of different tasks in representation space affect the performance of the coordinator?

---

> ### Author Response · Authors · 2025-11-21
> **Response to Reviewer Ev8j part 1**
>
> We thank the reviewer for the positive comments on our lightweight design, potential of application and high efficiency. We are also grateful for the constructive feedback to help us further improve the paper. We address the reviewer's concerns in the following. Please do not hesitate to let us know if you have any question.
>
> ---
>
> > **W1:** The effectiveness of the Trinity framework is limited compared to single-model baselines. As shown in Figure 3 and Table 1, compared with Gemini Pro 2.5, Trinity only brings an improvement about 1%-3% on all the in-distribution and out-of-distribution benchmarks except LiveCodeBench (where it outperforms GPT-5 for merely 3%). Considering the cost of training a coordinator and multi-turn interaction for selecting models, it shows limited effectiveness compared with simply selecting LLMs according to their adept tasks (e.g. select GPT-5 for coding tasks while Gemini Pro 2.5 for others).
>
> We agree that the coordinator is not “free” to train or deploy, especially given our limited compute budget. However, its cost is still small compared to training or even fine-tuning a modern LLM.
>
> Regarding effectiveness, it is important to look beyond per-task “1–3%” differences to the overall landscape. While on some individual benchmarks TRINITY’s improvement over the second-best method is numerically small, that second-best method is not consistent: it switches between different single models depending on the task. When we average across all in-distribution tasks, the best single-model configuration (Gemini Pro 2.5 with 5× context) is 2.84% worse than TRINITY. Moreover, some of these “small” absolute gains correspond to meaningful relative error reductions in the high-performance regime: for MATH500, where strong methods reach about 0.85–0.90 accuracy, TRINITY’s ≈2-point absolute improvement over the 2nd best method translates into an 11.76% relative error reduction, in addition to its gains on LiveCodeBench and other settings.
>
> In summary, TRINITY delivers improvements that are comparable in magnitude to those typically reported between successive generations of flagship LLMs on some tasks, without retraining any LLMs. Crucially, the framework offers a sustainable path forward. That is, as new LLMs are released, they can be plugged into the pool and immediately exploited by the coordinator setup, allowing users to track and benefit from rapid model progress without repeatedly paying enormous costs of training LLMs or hindered by the inaccessiblity of closed sourced LLMs.
>
> ---
>
> > **W2:** The necessity of LLM selection is not demonstrated in the ablation study. It is unclear whether the performance will degrade when the Tri-role selection process in the Trinity framework is applied on a fixed single-model baseline instead of a selected LLM.
>
> Following the reviewer's instruction, we have expanded the ablation study (Table 2 in the draft) to explicitly test settings where agent selection is disabled. Concretely, we fix the coordinator to always send queries to a single agent (Claude-4-Sonnet only, Gemini Pro 2.5 only, or GPT-5 only) while retaining the learned role-selection mechanism. For convenience, we reproduce the ablation table below.
>
> | Method                     | LiveCodeBench | MATH500   | MMLU      | RLPR      | Average   |
> | -------------------------- | ------------- | --------- | --------- | --------- | --------- |
> | TRINITY                    | **61.46**     | **88.00** | 91.56     | 40.72     | **70.44** |
> | Claude-4-Sonnet only       | 39.09         | 82.25     | 88.23     | 34.90     | 61.12     |
> | Gemini Pro 2.5 only        | 46.51         | 83.05     | 79.41     | **43.00** | 62.99     |
> | GPT-5 only                 | 59.54         | 75.66     | 90.74     | 37.87     | 65.95     |
>
> As the results show, removing agent selection and sending all queries to any single fixed agent leads to a substantial drop in average performance. The best single-agent variant (GPT-5 only) reaches 65.95, which is 4.49 points below TRINITY’s 70.44 average, and the other single-agent configurations fare even worse. This degradation occurs despite the fact that these variants retain the same role-selection mechanism, isolating the contribution of adaptive agent selection. Together, these findings demonstrate that the full TRINITY design, and in particular its learned agent selection component, is necessary to achieve the performance gains.

---

> ### Author Response · Authors · 2025-11-21
> **Response to Reviewer Ev8j part 2**
>
> > **Q1:** In Section 4.6, how does the separability of different tasks in representation space affect the performance of the coordinator?
>
> We thank the reviewer for raising this point. In theory, separability in the coordinator’s input space (the SLM hidden representations) is a necessary condition for a head to make meaningful agent and role decisions. To examine this connection more closely, we significantly extended Appendix A.3 by adding two additional experiments.
>
> First, a linear SVM trained on these representations achieves classification accuracies of 0.844@LiveCodeBench, 0.764@MATH500, 0.679@MMLU, and 0.544@RLPR. This ordering mirrors our empirical results: TRINITY shows the largest gains over baselines on LiveCodeBench and MATH500 and smaller gains on MMLU and RLPR, which may indicate that tasks with more separable representations in this space are easier to coordinate effectively.
>
> Then, to isolate separability effects from task-specific confounders, we further introduces a controlled synthetic experiment. We generate datasets that match the coordinator’s representation dimensionality and class structure (1024 dimensions, 7 agents, 4 task clusters), and then systematically vary separability by scaling inter-class distances while keeping within-class covariance fixed. We train the same head (*linear*) used in TRINITY on these synthetic datasets and observe a strong positive correlation between the separability index (between-class variance / total variance) and test accuracy, as shown in Figure 13.
>
> Together, the real-task SVM analysis and the synthetic study indicate that higher intrinsic separability in the representation space enables better coordinator performance, and that sufficiently separable task representations are a critical prerequisite for TRINITY to realize its performance.

---

### Official Review · Reviewer_3SKS · 2025-10-31

**Soundness:** 2
**Presentation:** 3
**Contribution:** 2
**Rating:** 4
**Confidence:** 3

**Summary:**

This paper introduces TRINITY, a novel framework for coordinating multiple, diverse foundation models (LLMs) at test-time. Instead of merging models at the weight level or using a large model for orchestration, TRINITY employs a highly lightweight coordinator. This coordinator consists of a small language model (SLM) (approx. 0.6B parameters) and an extremely small trainable head. The core mechanism involves a multi-turn protocol where the coordinator reads the full conversation history, extracts a contextual representation from the SLM's hidden state (specifically, the penultimate token's) , and then makes two decisions: (1) which LLM from the pool to select, and (2) which of three predefined roles—Thinker, Worker, or Verifier—that LLM should perform. A key technical contribution is the training methodology. The authors posit that standard reinforcement learning (RL) is ill-suited for this high-dimensional, budget-constrained optimization task. Instead, they successfully optimize the coordinator's lightweight head using a derivative-free evolutionary strategy, separable CMA-ES (sep-CMA-ES). The authors demonstrate that this approach is highly effective, achieving state-of-the-art (SOTA) performance on multiple benchmarks and setting a new record of 86.2% pass@1 on LiveCodeBench.

**Strengths:**

1. The paper's empirical results are its strongest asset. TRINITY consistently outperforms all individual baseline models (including powerful ones like GPT-5 and Gemini-2.5-Pro) and existing multi-agent routing methods across a wide range of tasks.
2. The paper's central claim—that a tiny coordinator with fewer than 20K learnable parameters  can effectively orchestrate a pool of powerful LLMs—is a significant and non-obvious finding.

**Weaknesses:**

1. The paper’s central claim of the "lightweight" coordinator's efficacy is insufficiently supported because it lacks comparisons to two of the most critical alternative coordinator designs. The authors never compare TRINITY to a system where a powerful LLM (e.g., GPT-4 or Gemini-2.5-Pro) is prompted to act as the coordinator. Such a baseline would use prompting to select the next agent and role. Its omission leaves a key question unanswered: is the 0.6B SLM's lightweight nature a genuine advantage (proving coordination is a simple task), or is it a limitation (acting as an understanding bottleneck that a stronger coordinator could overcome)? The paper’s comparison of training algorithms (ES vs. RL vs. RS)  is incomplete. A more standard and powerful baseline for training a router is SFT via behavioral cloning. The authors could have used their "Per-Question-Best" oracle data  to generate a dataset of expert decisions and then trained the 10K head. By omitting SFT, the claim that sep-CMA-ES is the superior optimization choice for this problem is not fully substantiated

2. A significant weakness lies in the rigid design of the "tri-role" protocol (Thinker, Worker, Verifier). These three roles are not dynamic but are hard-coded into the coordinator's head architecture, which has a fixed output dimension. This design severely limits the system's flexibility. As the authors note in their conclusion, the system cannot yet act on plans involving tools. To add a new, necessary role like a "Tool-User" or "Code-Executor," one cannot simply use a prompt; one would have to modify the model architecture and retrain the entire system from scratch. This makes the framework difficult to adapt to new capabilities or complex problems that do not fit neatly into the predefined tri-role schema.

**Questions:**

see my comments on Weaknesses

---

> ### Author Response · Authors · 2025-11-21
> **Response to Reviewer 3SKS part 1**
>
> We thank the reviewer for acknowledging our strong empirical results and that our central claim is "a significant and non-obvious finding". We are also grateful for the constructive feedback to help us further improve the paper. We address the reviewer's concerns in the following. Please do not hesitate to let us know if you have any question.
>
> ---
>
> > **W1-1:** The authors never compare TRINITY to a system where a powerful LLM (e.g., GPT-4 or Gemini-2.5-Pro) is prompted to act as the coordinator
>
> Following the reviewer's suggestion, we have included it in our experiments, where Gemini-2.5 Pro is prompted to act as a coordinator, with the same coordination settings (i.e., choosing a model and a role for each turn).
>
> The result shows that this approach underperforms TRINITY's trained coordinator (64.14 vs 70.44 in terms of average scores). We find that Gemini-2.5 pro struggles to comprehend and manage the properties of all 7 agents and its selections tend to be inconsistent and suboptimal. For completeness, we have also included the performance of the 7 agents' in the table. This suggests that prompting with closed-sourced LLMs is insufficient for capturing agents' inherent characteristics, which our trained coordinator acquires through training. See Table below for detailed results.
>
> | Method | Math500 | MMLU | RLPR | LiveCodeBench| Avg |
> |--------|---------|------|------|------------------|------|
> | TRINITY | 88.00 | 91.56 | 40.72 | 61.49 | 70.44 |
> | Gemini 2.5 pro as Coordinator | 78.67 | 83.26 | 26.83 | 26.28 | 53.76 |
> | Gemini Pro 2.5  | 85.30 | 91.53 | 39.57 | 40.14 | 64.14 |
> | Claude Sonnet 4 | 82.90 | 90.66 | 32.60 | 38.00 | 61.04 |
> | GPT 5  | 74.45 | 89.79 | 33.13 | 57.50 | 63.72 |
> | DeepSeek-R1-Distill-Qwen-32B | 78.50 | 84.41 | 32.75 | 0.14 | 48.95 |
> | gemma-3-27b-it | 56.00 | 63.58 | 14.93 | 7.21 | 35.43 |
> | Qwen3-32B (reasoning) | 76.85 | 83.28 | 34.35 | 31.21 | 56.42 |
> | Qwen3-32B (direct) | 73.15 | 84.02 | 30.60 | 26.79 | 53.64 |
>
> ---
>
> > **W1-2:** A more standard and powerful baseline for training a router is SFT via behavioral cloning.
>
> In response to the reviewer’s suggestion, we have added a dedicated appendix section (Appendix A.2) that evaluates supervised fine-tuning (SFT, i.e., behavioral cloning) as a baseline for training the coordinator. Labels are derived from the per-question-best oracle: for each question and seed, we select the LLM with highest reward, aggregate these choices across seeds via majority vote, and break ties by uniform sampling. We then train only the linear head on top of the frozen SLM using Adam, tuning over learning rates and batch sizes, and ultimately selecting a learning rate of $1\times10^{-6}$ and batch size 64.
>
> Empirically, this SFT baseline is competitive and non-trivial: the trained coordinator attains scores of 0.592 (LiveCodeBench), 0.786 (MATH500), 0.906 (MMLU), and 0.360 (RLPR), outperforming REINFORCE (RL) and random search (RS) baselines, but still consistently underperforming sep-CMA-ES across all tasks. For convienience, the table below compares the performance among different training methods.
>
> | Method      | LiveCodeBench | MATH500 | MMLU  | RLPR  |
> |------------|---------------|---------|-------|-------|
> | REINFORCE  | 0.253         | 0.459   | 0.500 | 0.266 |
> | RS         | 0.374         | 0.794   | 0.897 | 0.345 |
> | SFT        | 0.592         | 0.786   | 0.906 | 0.360 |
> | **sep-CMA-ES** | **0.615** | **0.880** | **0.916** | **0.401** |
>
> Thus, even when given access to strong oracle labels, SFT does not close the gap to sep-CMA-ES. This is consistent with our block–$\varepsilon$–separability analysis: SFT can exploit the coarse separability structure captured by an oracle-derived policy, but it remains constrained by label noise and finite-sample effects, whereas sep-CMA-ES directly optimizes in representation space and can better exploit the underlying geometry without requiring explicit action labels.
>
> Moreover, the main drawback of SFT in our setting is not performance but scalability in label generation. In the single-step setting studied here, constructing the oracle labels requires evaluating 7 agents over 7k datapoints with 3 seeds, i.e., $3 \times 7000 \times 7 = 147,000$ LLM queries. Extending this to multi-turn coordination increases the number of LLM calls exponentially under our experimental configuration. In contrast, sep-CMA-ES (and other label-free methods such as REINFORCE and RS) operate directly on task rewards and avoid explicit oracle label construction, making them far more practical for training scalable multi-turn coordinators despite SFT being a strong single-step baseline.

---

> ### Author Response · Authors · 2025-11-21
> **Response to Reviewer 3SKS part 2**
>
> > **W2:** To add a new, necessary role like a "Tool-User" or "Code-Executor," one cannot simply use a prompt; one would have to modify the model architecture and retrain the entire system from scratch.
>
> We thank the reviewer for this insightful comment regarding the extensibility of the tri-role schema, and we appreciate the opportunity to clarify how TRINITY accommondates new capabilities like tool-use without requiring architectural changes.
>
> - Role represents **workflow stages**, not capabilities: We wish to clarify that the 3 roles are designed as abstract meta-categories of the reasoning process (i.e., planning, execution and evalution), rather than specific functional capabilities. As defined in Section 3.2, the Worker role acts to "make concrete progress" and produce "actionable content", a "Tool-User" or "Code-Executor" is conceptually a Worker. Therefore, adding tool capabilies does not require a new architectural output (a 4th logit), it only requires the coordinator to assign the Worker role to an agent capable of tool manipulation. We validate this through zero-shot transfer evaluation on Tau-bench[1], a well-established agentic tool-use benchmark. In the maximum output setting, TRINITY achieves a pass@1 score of 0.62 on the airline subset, matching the GPT-5 (0.625).
> - Clarification on the conclusion's limitation statement: The limitation we noted regarding the gap between "abstract reasoning and grounded execution" refers to our experimental setup, not the coordinator's architecture. Specifically, our current agent pool did not include agents equipped with external tool APIs. To bridge this gap, one does not need to retrain the cooridinator, and simply needs to introduce a tool-equipped agent to the pool and utilize the Message Processsing module (i.e., prompting) to instruct it to use tools when the Worker role is assigned.
>
> [1] Shunyu Yao, Noah Shinn, Pedram Razavi, and Karthik Narasimhan.
> τ-bench: A benchmark for tool-agent-user interaction in real-world domains.
> In International Conference on Learning Representations (ICLR), 2025.

---

### Official Review · Reviewer_vdHP · 2025-11-01

**Soundness:** 3
**Presentation:** 3
**Contribution:** 3
**Rating:** 6
**Confidence:** 2

**Summary:**

The paper proposes TRINITY, which uses a lightweight coordinator to orchestrate multiple diverse LLMs at test time without modifying their weights. The coordinator consists of a compact SLM (≈0.6B) and a small decision head, and trained via sep-CMA-ES (diagonal covariance CMA-ES). Experiments on in-domain and out-domain benchmarks show consistent gains over single-model and routing/scaffolding baselines. And the ablation studies support the design choices of TRINITY's modules.

**Strengths:**

1. Motivation and Practicality: If substantiated, TRINITY offers a practical path to leverage strong closed and open models without retraining, with promising performance-to-cost potential. Additionally, the tri-role design and hidden-state routing signal could influence future multi-model systems.

2. Originality: TRINITY leverages the hidden states of small LMs to capture the optimal models and roles for current inference states. Specifically, the explicit tri-role protocol is a clean and practical abstraction. Using penultimate-token hidden states (instead of generated text) to drive coordination decisions is also a thoughtful and effective (revealed by the ablations) design. And the choice of sep-CMA-ES and its theoretical explanations about block-ε-separability perspective are interesting optimization angles for noisy binary rewards and high evaluation cost.

3. Clarity: The paper is well-written and easy to follow. Specifically, the method is well-defined, with clear problem formulation and a compact parametrization. The multi-turn protocol and role-specific prompting are coherent. Ablations and representation separability analyses support the claim that a linear head over SLM hidden states can be effective.

**Weaknesses:**

The main weaknesses are in baseline settings. Addressing these would substantially strengthen the paper.

1. The paper sets per-call max generation (4096 tokens) and turn limit (K=5) but does not report actual token usage of TRINITY and other baselines.
2. See the left three baselines (Gemini2.5 / GPT-5 / Claude) in Figure 3. They are set to generate 4k or 20k within a SINGLE turn while TRNITY adopts multi-turn designs (thinker, worker, verifier). And any parallel TTS techniques are not considered as well. I think it is necessary to evaluating simple parallel strategies (like major@5) or more complicated ones towards a more fair comparison.

**Questions:**

Please see the weakness above.

---

> ### Author Response · Authors · 2025-11-21
> **Response to Reviewer vdHP**
>
> We thank the reviewer for the positive comments on our paper's motivation, practicality, originality and clarify. We are also grateful for the constructive feedback to help us further improve the paper. We address the reviewer's concerns in the following. Please do not hesitate to let us know if you have any question.
>
> ---
>
> > **W1:** The paper sets per-call max generation (4096 tokens) and turn limit (K=5) but does not report actual token usage of TRINITY and other baselines.
>
> We have added token usage tables for all baseline methods in the revised appendix (also see tables below), and highlighted the key findings in the main text.
>
> Compared to other coordination methods in Table 1, TRINITY uses fewer tokens than most coordination methods (MOA, Smoothie, MASRouter). RouterDC is an exception due to its one-turn coordination nature, but that also significantly limits its performance.
>
> **Table1: Average number of output tokens. Agent Coordination Baselines**
> | Model | Math500 | MMLU | RLPR | LiveCodeBench|
> |-------|---------|------|------|---------------|
> | TRINITY | 2,853 | 1,200 | 2,141 | 1,999 |
> | MOA | 6,871 | 5,218 | 11,086 | 21,634 |
> | RouterDC | 624 | 374 | 811 | 1,552 |
> | Smoothie | 6,472 | 4,718 | 10,580 | 17,864 |
> | MASRouter | 4,260 | 1,847 | 5,370 | 8,401 |
>
> We also include token usage for all single model configurations in Table 2 to 4: 5x self-reflection, 5x context (20,480 tokens), and default context (4,096 tokens). One interesting finding is that the actual token costs for 5x context closely match default context across most models, except for Gemini on LiveCodeBench. This explains their similar performance in Figure 2 in the paper, with only Gemini showing gains from extended context.
>
> **Table 2: Average number of output tokens. 5× Self-Reflection**
> | Model | Math500 | MMLU | RLPR | LiveCodeBench|
> |-------|---------|------|------|---------------|
> | Qwen3-32B (direct) | 2,075 | 1,746 | 1,949 | 4,207 |
> | Qwen3-32B (reasoning) | 2,692 | 2,213 | 3,349 | 7,575 |
> | DeepSeek-R1-Distill-32B | 3,988 | 3,811 | 4,609 | 12,228 |
> | Gemma-3-27B | 1,704 | 820 | 1,750 | 3,714 |
> | Claude Sonnet 4 | 1,834 | 1,293 | 1,580 | 3,210 |
> | GPT-5 | 577 | 428 | 895 | 1,971 |
> | Gemini-2.5-Pro | 5,142 | 5,460 | 6,710 | 11,046 |
>
> **Table 3: Average number of output tokens. 5× Context Setting**
> | Model | Math500 | MMLU | RLPR | LiveCodeBench|
> |-------|---------|------|------|---------------|
> | Qwen3-32B (direct) | 447 | 152 | 392 | 192 |
> | Qwen3-32B (reasoning) | 1,019 | 382 | 1,047 | 1,784 |
> | DeepSeek-R1-Distill-32B | 1,343 | 538 | 1,369 | 4,066 |
> | Gemma-3-27B | 342 | 146 | 336 | 159 |
> | Claude Sonnet 4 | 367 | 218 | 300 | 518 |
> | GPT-5 | 221 | 66 | 219 | 1,207 |
> | Gemini-2.5-Pro | 1,153 | 579 | 787 | 5,753 |
>
> **Table 4: Average number of output tokens. Default Context Setting (4096 tokens)**
> | Model | Math500 | MMLU | RLPR | LiveCodeBench|
> |-------|---------|------|------|---------------|
> | Qwen3-32B (direct) | 521 | 154 | 406 | 419 |
> | Qwen3-32B (reasoning) | 995 | 397 | 1,191 | 1,789 |
> | DeepSeek-R1-Distill-32B | 1,175 | 485 | 1,181 | 3,443 |
> | Gemma-3-27B | 437 | 147 | 330 | 483 |
> | Claude Sonnet 4 | 382 | 217 | 304 | 530 |
> | GPT-5 | 218 | 66 | 220 | 1,113 |
> | Gemini-2.5-Pro | 819 | 578 | 774 | 2,396 |
>
> ---
>
> > **W2:**  I think it is necessary to evaluating simple parallel strategies (like major@5) or more complicated ones towards a more fair comparison.
>
> We have included the majority voting@5 results in Section 4.1 and Appendix 7.3 of the paper. Here is the table that summarizes the result:
>
> **Majority@5 baseline results for MMLU**
>
> | Model | Avg Score |
> |-------|-----------|
> | Gemini Pro 2.5 | **91.57** ± 0.70 |
> | GPT-5 | 91.31 ± 0.23 |
> | Claude-4-Sonnet | 90.99 ± 0.39 |
>
> Moreover, while TRINITY is designed to handle a broad spectrum of tasks, majority voting@5 and similar parallel TTS approaches have significant limitations. Majority voting only applies to tasks with a small, discrete set of candidate outputs, such as multiple-choice benchmarks like MMLU. Best-of-N and tree search methods, while technically applicable to tasks like coding, require repeated access to ground truth evaluators and select among candidates, which may not be feasible in the real-world setting and our experiments do not assume access to such evaluation oracles.
>
> Although Gemini Pro 2.5 with majority voting@5 reaches similar performance as TRINITY in MMLU, a task which is not TRINITY's strong suit, TRINITY is applicable for a much wider set of tasks, as compared with majority voting@5. Also, some existing methods that are compared in Figure 3 involve parallel strategies. For example, Mixture of Agent (MoA) samples in parallel from all worker models and uses the summarized answer as the final response, which can be considered a generalized version of majority voting. Overall, these results suggest TRINITY's advantages over parallel strategies.

---

### Author Response · Authors · 2025-11-21
**Message to all reviewers and AC**

We thank the reviewers for their insightful and constructive feedback. Your comments have allowed us to clarify ambiguities and further improve the paper. We have revised our paper to reflect these clarifications and improvements, with all changes highlighted in blue for your convenience. The following is a summary of our revision.

* For **reviewer vdHP** we have
    1. Added detailed token-usage tables for TRINITY, all collaboration baselines, and all single-model configurations under 5× self-reflection, 5× context, and default context, and discussed how TRINITY is typically more token-efficient than other agents coordination methods while remaining comparable to strong single models;
    2. Incorporated parallel-strategy baselines, in particular majority-vote@5 on MMLU, and clarified their scope by showing that while such methods can be strong on multiple-choice benchmarks, TRINITY remains competitive there and, importantly, continues to apply effectively to open-ended coding, math, and reasoning tasks where simple parallel voting is not directly applicable.

* For **reviewer 3SKS** we have
    1. Implemented an “LLM-as-coordinator” baseline where Gemini-2.5-Pro is prompted to choose both model and role each turn, and found that this prompted coordinator underperforms the trained TRINITY coordinator and even some individual agents, suggesting that prompting alone is insufficient to robustly learn agent characteristics;
    2. Added a supervised fine-tuning / behavioral cloning baseline trained on oracle agent selection labels, and shown that although SFT is competitive and stronger than REINFORCE and random search, sep-CMA-ES consistently performs best and avoids the prohibitive label-generation cost inherent to SFT in multi-turn coordination;
    3. Clarified that our tri-role scheme encodes workflow stages (Thinker/Worker/Verifier) rather than hardcoded capabilities, so new skills such as tool use or code execution can be introduced by adding tool-enabled agents and prompting them appropriately under the Worker role without changing the architecture.

* For **reviewer Ev8j** we have
    1. Quantified the cost–effectiveness of training the coordinator and shown that, averaged over in-distribution tasks, TRINITY achieves non-trivial gains over the best single-model setups, including substantial relative error reductions in the high-accuracy regime (e.g., 11.76% on MATH500);
    2. Added ablations that disable agent selection by forcing all queries to a fixed LLM (Claude-only, Gemini-only, or GPT-5–only) while retaining role selection, which consistently lowers average performance and isolates the benefit of adaptive agent coordination;
    3. Significantly expanded the analysis of representation space separability with per-task linear SVM accuracies and a controlled synthetic study, showing a positive correlation between separability in the SLM’s representation space and the coordinator’s performance.

* For **reviewer bpif** we have
    1. Improved organization and clarity by refining Section 2’s notation, making the penultimate-token head input explicit in Figure 2, and restoring the key empirical analysis to the main text with a clearer explanation of our choice of sep-CMA-ES;
    2. Clarified the coordinator parametrization by distinguishing singular value fine-tuning on selected backbone layers from full training of the head, and added a parameter-count table to quantify how lightweight each component is;
    3. Expanded the tri-role ablations, including two-role and no-tri-role variants, showing that merging or removing roles reduces performance and highlighting the importance of a dedicated Thinker and the full tri-role protocol;
    4. Added a supervised (state, action) SFT baseline based on oracle per-question-best labels, comparing it to sep-CMA-ES, REINFORCE, and random search, and discussing why SFT is both slightly weaker and much less scalable in the multi-turn setting due to label-generation cost.

---

### Meta-Review · Area_Chair_Af6x · 2025-12-21

**Summary:**

All reviewers commented on the weakness of the baselines. Thus, during rebuttal, various such baselines were provided (no orchestrator, off-the shelf orchestrator etc.). It hence seems most weaknesses of this paper were addressed, leaving this paper with perhaps medium excitement but agreement on soundness.

**Reviewer Concerns:**

Most issues, including writing (which is likely only partially addressed) and baselines (many of which were added and the results seem ok. Of course, no one could have reverified the settings).

**Reviewer Scores:**

That is not a fair, relevant or meaningful question. I protest the way this was all handled.
A Reviewers are not here, and ToM is weak, at least mine and the one literature study. I will not try to predict people.
B Scores are, anyway, a weak signal of interest; a paper should not be accepted or rejected just based on it. An AC's job is to look at the specific weaknesses and translate them into a recommendation.
C There are about 100 pages of discussions for me to read overall, in addition to the discussions I monitored and were just replaced, this is beyond my personal ability to do fairly. I did my best effort.


One of the reviews was misplaced, but they can't correct it now...

---

### Decision · Program_Chairs · 2026-01-26

Accept (Poster)